

# Towards an ideal water-energy-food nexus model: moving beyond silos to integrated resource governance

Wei Li[1,2], Philip J. Ward[3,4], Lia van Wesenbeeck[1,2]

[1]School of Business and Economics, Vrije Universiteit Amsterdam, De Boelelaan 1105, 1081 HV, Amsterdam, The Netherlands
[2]Amsterdam Centre for World Food Studies, Vrije Universiteit Amsterdam, De Boelelaan 1105, 1081 HV, Amsterdam, The Netherlands
[3]Institute for Environmental Studies, Vrije Universiteit Amsterdam, De Boelelaan 1105, 1081 HV, Amsterdam, The Netherlands
[4]Deltares, Boussinesqweg 1, 2629 HV, Delft, The Netherlands

*Correspondence to*: Wei Li (w.li2@vu.nl)

**Abstract.** The water-energy-food (WEF) nexus applies systems thinking to transcend siloed sectoral perspectives and foster integrated resource governance. This study identifies six key objectives that an ideal model for the WEF nexus should achieve: ensuring resource security; promoting resource circularity; enabling transferability across spatial and temporal scales and geographic scopes; facilitating comprehensive identification and quantification of resource interactions; integrating economic, environmental, and societal considerations; and ensuring theoretical rigor and empirical solvability. No existing WEF nexus model simultaneously fulfils all six objectives. To address this gap, we develop the first transparent and comprehensive WEF nexus model that achieves all six objectives. The proposed model links water quantity and quality —including pollutants and temperature—with energy and food systems to analyse system-wide water dynamics. The model explicitly represents water-energy interactions, capturing how thermoelectric generation alters water thermal regimes and, in turn, affects hydrological processes. It also accounts for human-water interactions by incorporating return flows of water after human consumption, emphasizing water circularity. The model's applicability is illustrated through an example of the Beijing-Tianjin-Hebei region in China, and its broader empirical and policy relevance is demonstrated through a set of potential scenarios. These advances provide a systems foundation for understanding hydrological science and for developing sustainable, efficient, and equitable resource strategies.

## 1 Introduction

The water-energy-food (WEF) nexus aims to capture the interdependence between water, energy and food in a holistic approach (Schlör et al., 2021). Moving beyond traditional sector-specific silos, the WEF nexus is recognized as a promising system approach to advance sustainable development (European Commission. Joint Research Centre. et al., 2021). Despite the surge in WEF nexus research since 2011, the gap between theoretical frameworks and empirical applications remains wide (Li et al., 2025). This poses the risk of losing the holistic and integrated perspective of the nexus approach (Correa-



Cano et al., 2022). Derived from the initial conceptualization of the WEF nexus and research gaps identified in the extant literature, an ideal model for the WEF nexus seeks to achieve at least six objectives simultaneously.

First, an ideal model for the WEF nexus should achieve resource security by addressing both the quality and quantity aspects

of resources (Hoff, 2011; World Economic Forum, 2011). Within WEF nexus research, the quantification of resources is relatively well-established, while the quality assessment has gained much less attention (Heal et al., 2021; Yang et al., 2016). WEF nexus studies that include quality aspects focus on water quality, neglecting quality concerns in food (Mortensen et al., 2016; Ramirez et al., 2021). In addition, water quality is assessed by a limited range of pollutants (Liang et al., 2019). Specialized water models have integrated water temperature (Van Vliet et al., 2021) and heavy metals (Zhou et al., 2023)

especially in rivers, but these are absent in WEF nexus studies.

Second, an ideal model for the WEF nexus should consider the return flows of water and food after human consumption, adhering to resource circularity (European Commission. Joint Research Centre., 2019). This is a logical consideration that has so far been largely absent in WEF nexus studies grounded in microeconomic approaches, where human consumption marks the end of resource flows. In the interdisciplinary context of the WEF nexus, this risks missing crucial details

concerning interconnected resources (Vahedi et al., 2025).

Third, an ideal model for the WEF nexus should be transferable across different spatial and temporal scales, as well as geographic contexts (Naidoo et al., 2021). However, many WEF nexus models have limited real-world applicability (Taguta et al., 2022), while some are tailored for specific regions or research scales (Albrecht et al., 2018).

Fourth, an ideal model for the WEF nexus should comprehensively identify and quantify the three self-interactions of water,

energy and food resources, and the six mutual interactions between them (Daher and Mohtar, 2015; Ramos et al., 2020; Shivakumar et al., 2021). Yet most models overlook resource self-interactions and fail to comprehensively capture resource interactions (Albrecht et al., 2018; Cremades et al., 2021; Taguta et al., 2022).

Fifth, an ideal model for the WEF nexus should integrate economic, environmental, and societal considerations (Bazzana et al., 2021; Hoff, 2011). Most models focus on economic and environmental considerations, with less emphasis on societal

concerns (Zhang et al., 2024). Environmental considerations include carbon emissions (Yue et al., 2022), water pollution (Li et al., 2019), and ecosystem services (Yue et al., 2021). Furthermore, these considerations are often treated separately, as in multi-objective models (Alherbawi et al., 2021; Yue et al., 2022).

Finally, an ideal model for the WEF nexus should ensure both theoretical rigor and empirical solvability. Empirical studies typically present only the final results, while rigorous theoretical proofs remain rare. An ideal WEF nexus model requires

theoretical rigor as a prerequisite and should also be empirically solvable, ensuring efficient solutions within a reasonable time frame on standard computers.

To date, no WEF nexus model has simultaneously addressed all six objectives. This study addresses this gap by developing a pioneering model that achieves all six. Expanding a comprehensive regional water economy model for the Jordan River Basin (hereafter the "JRB model"), this study progressively extends its scope to construct a transparent and comprehensive





WEF nexus model. The model's applicability is demonstrated through an example of the Beijing-Tianjin-Hebei (abbreviated as BTH) region in China, and its broader empirical and policy relevance is demonstrated through a set of potential scenarios. The novelty of this work stems from three main contributions. First, it advances hydrological assessment by integrating water quantity and quality, including both soluble and insoluble pollutants as well as water temperature, providing a more comprehensive representation of hydrological system dynamics. Second, it explicitly models water-energy interactions,

showing how thermoelectric generation using cooling water modifies thermal regimes and subsequently influences hydrological processes and water availability. Third, it incorporates human-water interactions by accounting for return flows of water after human consumption, highlighting water circularity. Collectively, these innovations provide a novel systems model for advancing hydrological science and for developing sustainable, efficient, and equitable resource management strategies.

## 75  2 Water balance model from the JRB model

The theoretical foundation of the JRB model originates from the internal work "Optimal calibration and control in a regional water economy model for the Jordan River Basin" by M.A. Keyzer (2015), which is available upon request. Theoretically, the JRB model adopts welfare optimization as the economic framework, with hydrology as a central component of the production technology.

Here, an explicit description of the water balance model from the JRB model is presented. In brief, the water balance model consists of a series of interconnected equations, each delineating a mass balance between water origins and water uses, for a specific time step, location, water quality, and soil layer. Water also flows across these four dimensions, thus fostering tight interconnections among all equations.

Specifically, in the water balance model, the time step is denoted by $t$ ($t = 1, ..., T$), spanning the annual hydrological cycle.

The research region is partitioned into $S$ polygons, fully covering the area without any remainder, with each polygon representing a distinct location denoted by $s$ ($s = 1, ..., S$). Considering natural water flows (both horizontally and vertically) and anthropogenic activities related to water, the soil layers, denoted by $l$ ($l = 1, ..., L$), are classified into: land surface layer, river layer, anthropogenic activity layer, root zone layer, and aquifer layer. A detailed introduction to each soil layer and their interactions is provided in Appendix A.

The final dimension to consider is water quality, which varies over time $t$, across locations $s$, and through different layers $l$. In the water balance model, water quality is denoted by $h$ ($h = 1, ..., H$). Based on local contexts, water quality in the JRB model is assessed using three soluble pollutants: biochemical oxygen demand, salinity, and nitrate. Instead of using a set of attributes to represent water quality in each time, location, and soil layer, soluble pollutants are treated as flows, for which mass balances must hold, as for pure water. Specifically, each soluble pollutant is represented as a distinct flow with its own

saturation concentration. Observed concentrations are obtained by mixing the volumes of these flows with the volume of pure water. Given that soluble pollutants flow with water and cannot be separated, the water balance model integrates all





flows, including both pure water flow and flows containing soluble pollutants. This is reflected in the model, where the water quality at a specific combination of $(s, t, l)$ is represented as a given blend of qualities $h$.

For every combination of quality $h$, location $s$, time $t$, and soil layer $l$, a water stock $x$ is defined. It is assumed that water

adheres to the material balance with non-negative stocks (i.e., $x \geq 0$) and flows under a discrete representation in location and time. To represent fluidity, the water flow is contiguous across location and time (no jumps).

Water can be allocated to human water use $d^i$ of demand type $i$ ($i = 1, \dots, I$), water inputs $e^j$ of anthropogenic project $j$ ($j = 1, \dots, J$) and natural outflow $\tilde{x}$ (including retention, represented by water of the same quality $h$ remaining at the same location $s$ and soil layer $\ell$ from time $t$ to $t + 1$). These three water uses are denoted as vectors of length $H * S * T * L$, and

are shown as fixed fractions of the total available water stock $x$:

$$d^i = \Delta^i x \tag{1}$$

$$e^j = H^j x \tag{2}$$

$$\tilde{x} = Rx \tag{3}$$

Where $\Delta^i$ denotes the diagonal matrix with fixed fractions $\delta^i$, $H^j$ denotes the diagonal matrix with fixed fractions $\eta^j$, and $R$

denotes the diagonal matrix with fixed fractions $\rho$. These three coefficients are all square matrices of order $H * S * T * L$, and are non-negative. Since retention is included as a component of natural outflows, these fractions must logically sum to one. This implies that the sum of these three matrices is an identity matrix of order $H * S * T * L$. We note that the fixed fractions will become endogenous in the embedding of the balances in the broader welfare framework.

The JRB model reflects the fact that the water user must take the blending quality of water as it comes rather than picking

the best qualities only. For them, the water quality flows from a single "tap" under their control, instead of multiple "taps" each containing only one specific pollutant. To represent this, water quality is treated as mixing volumes with standard pollutant concentrations, the transformation of which is related to anthropogenic production (e.g. wastewater treatment) or natural processes (e.g. evaporation). Therefore, $\delta^i$, $\eta^j$, and $\rho$ do not depend on index $h$, and Eqs. (1), (2), and (3) can be written as:

$$d^i_{hstl} = \delta^i_{stl} \, x_{hstl} \tag{1a}$$

$$e^j_{hstl} = \eta^j_{stl} \, x_{hstl} \tag{2a}$$

$$\tilde{x}_{hstl} = \rho_{stl} \, x_{hstl} \tag{3a}$$

Corresponding to the three water uses, the origins of the water stock consist of: natural inflows $A\tilde{x}$, where $A$ denotes the matrix of natural water flow coefficients; return flows from human resource use $B^{0i} d^i$, where $B^{0i}$ denotes the matrix of

return coefficients of demand type $i$; flows from anthropogenic projects $B^j e^j$, where $B^j$ denotes the matrix of coefficients of anthropogenic project $j$; and net exogenous availability $b$, which denotes the precipitation minus immediate evaporation and plus inflow from outside the system (if any). Here, $A$ is a square matrix of order $H * S * T * L$, $B^{0i}$ and $B^j$ are potentially square matrices of order $H * S * T * L$, and $b$ is a vector of length $H * S * T * L$.

To represent the natural water flows, **Assumption 1a** is introduced.





**Assumption 1a (Linear water outflows):** The water outflow function $v_{hstl}(x)$ from $h's't'l'$ into $hstl$ possesses the following properties:

   i. volume at standard temperature is unit of measurement;

   ii. linear form: $v_{hstl}(x) = \sum_{h's't'l'} A_{hstl,h's't'l'} x_{h's't'l'}$;

   iii. contiguity: flows from $s't'l'$ into adjacent $stl$ only;

   iv. non-negativity: $A_{hstl,h's't'l'} \geq 0$;

   v. no flows backward in time: $A_{hstl,h's't'l'} = 0$ for $t < t'$;

   vi. non-expansiveness (water does not breed): outflows from $h's't'l'$ do not exceed availability or fall below it due to evaporation and other losses, as expressed by the restriction on the column sums of matrix $A$: $\sum_{h's't'l'} A_{h's't'l',hstl} < 1$.

If **Assumption 1a** holds, the water flows can be represented through a linear equation with spatially and temporally connected water balances for each $(h, s, t, l)$, with the origins of water stock on the right and the uses of the available water stock on the left, as shown in Eq. (4):

$$\sum_i d^i + \sum_j e^j + \tilde{x} = A\tilde{x} + \sum_i B^{0i} d^i + \sum_j B^j e^j + b \tag{4}$$

Note that each variable and parameter in Eq. (4) carries the subscript of four dimensions: water quality, $h$; location, $s$; time $t$; and soil layer $l$. In the JRB model, each combination of $(h, s, t, l)$ defines a cell, which is the minimum unit. Each cell includes reasonable representations of essential features. Within each cell, a multitude of natural and anthropogenic activities occur between water origins and uses, while water concurrently flows across cells.

## 3 Resource balance model

The JRB model exclusively represents water. This study begins developing the resource balance model for the WEF nexus by integrating food and energy.

First, while water exists as a single commodity, the representation of food and energy requires multiple distinct commodities. Hence, a fifth dimension is added to the water balance model (shown in Eq. (4)), namely the commodity, indexed as $m = 1, \dots, M$. It is assumed that each commodity $m$ is homogeneous. Distinguishing commodities allows for the identification of self-interactions within each resource (i.e., interactions between commodities within each resource). Other commodities are aggregated into a single commodity class, referred to as "others".

Second, it is assumed that energy commodities and food commodities are exclusively concerned with human-made firm activities, without natural flows. Naturally occurring crops, wild fruit trees, and spontaneous vegetation that are not deliberately cultivated or maintained through human intervention are classified under an aggregated "others" category within firm activities, rather than being treated as natural flows. Therefore, the natural outflows $\tilde{x}$ and natural inflows $A\tilde{x}$ in Eq. (4) are not included for energy and food commodities. Besides, the JRB model uses the "anthropogenic project" to distinguish



human activities from natural water flows. In the WEF nexus model, "anthropogenic project" is renamed as "firm activity" to align with the commonly used terminology in production theory.

Third, with the integration of food commodities and energy commodities, the reinterpretation of human demand type $i$ and firm activity $j$ is needed. In the resource balance model, human resource demand includes both essential consumption (e.g., drinking, cooking) and functional usage (e.g., dish washing, toilet flushing, showering, laundry). To distinguish between human resource users, human demand is classified by type $i$. Firm activity $j$ covers all human-made activities distinct from natural processes, including crop farming, water harvesting, and energy loss.

So far, the resource balance model for the WEF nexus model has been formulated. The inclusion of food and energy does not change the appearance of Eq. (4), however, the subscript of each variable and parameter now has five dimensions, $(h, s, t, l, m)$. While several variables and parameters in the resource balance model may not be included for specific resources, they are retained for the generality across all three resources.

The form of the resource balance model is represented as Eq. (5).

$$\sum_i d^i + \sum_j e^j + \tilde{x} = A\tilde{x} + \sum_i B^{0i} d^i + \sum_j B^j e^j + b \tag{5}$$

Resource uses are on the left-hand side of Eq. (5), consisting of human resource use $d^i$ of demand type $i$ ($i = 1, \dots, I$), resource inputs $e^j$ in firm activity $j$ ($j = 1, \dots, J$) and natural water outflow $\tilde{x}$. Origins of the resource availability are on the right-hand side, consisting of natural water inflows $A\tilde{x}$, return flows from human resource use $B^{0i} d^i$, flows from firm activity $B^j e^j$ and net exogenous resource availability $b$. Eq. (5) shows the balance between the uses of the available resource stock and the origins of this stock for each $(h, s, t, l, m)$. Five dimensions result in a total of $H * S * T * L * M$ resource balances, which capture all resource interactions within the nexus.

Building on the established resource balance model, further modifications and reinterpretations are undertaken to fulfil **Objective 1**, **2**, and **3**.

### 3.1 Objective 1: include both quality and quantity aspects of resources

The resource balance model is based on mass balances, in which all stocks and flows of resources are measured in units of resource quantity. Having accounted for the quantity aspect of resources, this subsection addresses the quality evaluation of the three resources.

#### 3.1.1 Extend to a comprehensive water quality measurement

The JRB model measures water quality by three pollutants, i.e., biochemical oxygen demand, salinity, and nitrate. Expanding beyond this, this model aims for a more comprehensive evaluation of water quality, incorporating both soluble and insoluble water pollutants, as well as water temperature.

**Water pollutants**





The JRB model's treatment of soluble pollutants is adopted, where each pollutant is presented as a separate flow with its own maximum concentration. Thus, water flow in any stock or flow is a mixture of pollutant flows and pure water, resulting in a composite flow confronted by the water user.

The key distinction between pollutants in this model lies in their flow patterns, which are captured by the matrix of natural water flow coefficients $A$, in Eq. (5), while other properties are treated as homogenized. The five-dimensional subscript of $A$ tracks pollutant flows across cells, particularly locations $s$, time steps $t$, and soil layers $l$. Soluble pollutants dissolve in water and move with the pure water flow, sharing the same $A$, while insoluble pollutants are either suspended or settled in the water, moving independently with their own $A$. By not strictly classifying pollutants as either soluble or insoluble, this approach more accurately reflects real-world scenarios where such distinctions may not always apply.

**Water temperature**

Water temperature plays a crucial role by affecting the interactions between water and other resources, particularly water for energy and water for food. For instance, thermo-electric (nuclear and fossil-fuelled) power plants rely on cooling water to function efficiently (King et al., 2008). When river temperatures rise during hot summers, cooling water becomes scarce, significantly reducing energy generation (Behrens et al., 2017). Additionally, elevated water temperatures foster the proliferation of various bacteria (Calero Preciado et al., 2021), which degrades irrigation water quality and adversely affects crop growth (Assouline et al., 2015).

However, no WEF nexus model incorporates water temperature as a component of water quality measurement. To address these, water temperature is introduced as the sixth dimension, denoted as $c$, alongside $h, s, t, l,$ and $m$. It is assumed that water temperature is homogeneous within a given area $s$, soil layer $l$, and time step $t$. Two virtual water flows are defined: *blue water*, representing the lowest temperature of liquid water; and *red water*, representing the highest temperature of liquid water. The actual water temperature faced by water users is modelled as a blend of these two extreme virtual flows. It is important to note that these virtual water flows represent a characteristic of water, rather than a physical flow itself.

Through this approach, two issues are effectively addressed: (1) the natural fluctuations in water temperature, which are captured through cross-balance interactions across dimension $c$ and $t$; and (2) the interactions between water temperature and other pollutants, which is captured through cross-balance interactions across dimension $c$ and $h$. Moreover, by incorporating bacteria as a pollutant flow within the $h$ dimension, this model can quantify the impact of rising water temperatures on bacterial proliferation through the cross-balance interactions.

Given all these changes, **Assumption 1a** is modified to **Assumption 1**.

**Assumption 1 (Linear water outflows):** The water outflow function $v_{hstlc}(x)$ from $h's't'l'c'$ into $hstlc$ possesses the following properties:

    i. volume of blue water and red water is unit of measurement;

    ii. linear form: $v_{hstlc}(x) = \sum_{h's't'l'c'} A_{hstlc,h's't'l'c'} x_{h's't'l'c'}$;

    iii. contiguity: flows from $s't'l'$ into adjacent $stl$ only;



iv. non-negativity: $A_{hstlc,h's't'l'c'} \geq 0$;

v. no flows backward in time: $A_{hstlc,h's't'l'c'} = 0$ for $t < t'$;

vi. non-expansiveness (water does not breed): outflows from $h's't'l'c'$ do not exceed availability or fall below it due to evaporation and other losses, as expressed by the restriction on the column sums of matrix $A$: $\sum_{h's't'l'c'} A_{h's't'l'c',hstlc} < 1$.

Adopted form the JRB model, users must accept the blending quality of water as it comes, rather than selecting the best

quality only. This implies that the water user lacks any liberty of choosing the desired quality $h$ and temperature $c$, represented by the absence of subscript $h$ and $c$ in $\delta_{stl}^i$, $\eta_{stl}^j$, and $\rho_{stl}$ in Eqs. (1a), (2a) and (3a). This still holds for water in the resource balance equation for the WEF nexus model. The subscript $m$ is omitted here, as it specifically refers to the commodity water. For clarity, non-relevant subscripts are omitted throughout this study, provided there is no risk of confusion.

**3.1.2    Add food quality**

Instead of focusing on the physical, microbial, or chemical causes of spoilage, food quality is classified from the perspective of resource interactions. Specifically, three discrete food classes are distinguished: (1) human consumption quality, denoted as $h_1$; (2) feed quality for animals, denoted as $h_2$; and (3) energy conversion quality, denoted as $h_3$.

It is assumed that $h_1 > h_2 > h_3$ always holds. In line with practical operations, the food commodity of higher quality can be

used for lower-quality purposes, but not vice versa. Moreover, the quality of food commodities deteriorates over time through storage, transportation, and other processes, resulting in a transition between quality classes. This model can explicitly capture this degradation process.

Users must accept the blending quality of water as it is, whereas for food commodities with observable quality, they can visually assess the quality and make consumption decisions accordingly. To address unobservable quality differences in

food, it is further assumed the existence of a food quality rating agency. Supported by this agency, all users make consumption decisions based on the actual quality of food commodities.

For food commodities, there are no natural flows; all flows are human made. Therefore, natural outflows are not needed, as denoted in Eq. (3). Food can be allocated to human use $d^i$ of demand type $i$ ($i = 1, \dots, I$) and food inputs $e^j$ of firm activity $j$ ($j = 1, \dots, J$). These two uses are represented as fixed fractions of the total available food stock $x$, as described in the Eqs.

(1) and (2).

Unlike water, users retain the liberty to consume food commodities according to their intended purpose, rather than consuming any quantity of food irrespective of its quality. To reflect this, Eqs. (1) and (2) are rewritten as:

$$d_{hstlm}^i = \delta_{hstlm}^i \, x_{hstlm} \qquad (1b)$$





$$e^j_{hstlm} = \eta^j_{hstlm} \; x_{hst\ell m} \tag{2b}$$

Where $\delta^i$ and $\eta^j$ represent fixed fractions of the diagonal matrices $\Delta^i$ and $H^j$, respectively. Both $\Delta^i$ and $H^j$ are square matrices of order $H * S * T * L * M$. They are non-negative, and their sum must logically equal one.

The food for these uses is produced using the conventional economic production function, which satisfies **Assumption 2**.

**Assumption 2 (Linear food production function):** The food production function $f_{hstm}(x)$ for firm activity $j$ transforms inputs $e^j_{h's't'm'}$ from $h's't'm'$ into outputs of $hstm$ with the following properties:

i. linear form: $f_{hstlm}(x) = \sum_{h's't'm'} B^j_{hstm,h's't'm'} \, e^j_{h's't'm'}$

    ii. non-negativity: $B^j_{hstm,h's't'm'} e^j_{h's't'm'} \geq 0$;

    iii. no flows backward in time: $B^j_{hstm,h's't'm'} = 0$ for $t < t'$;

    iv. non-expansiveness (food production does not exceed available resources): $\sum_{h's't'm'} B^j_{h's't'm',hstm} < 1$.

### 3.1.3 Homogeneous energy quality

Consistent with the treatment of energy quality in related models (Benchekroun et al., 2023), energy is assumed to be homogenous. Energy commodities have no natural flows. Therefore, Eqs. (1) and (2) apply to energy commodities, without the need for Eq. (3). Due to their homogeneous quality, Eqs. (1) and (2) are rewritten as Eqs. (1a) and (2a). Energy is also produced using the conventional production function, which satisfies **Assumption 3**.

**Assumption 3 (Linear energy production function):** the energy production function $f_{stm}(x)$ for firm $j$ transforms inputs 270 $e^j_{s't'm'}$ from $s't'm'$ into outputs of $stm$ with the following properties:

    i. linear form: $f_{stm}(x) = \sum_{s't'm'} B^j_{stm,s't'm'} \, e^j_{s't'm'}$;

    ii. non-negativity: $B^j_{stm,s't'm'} e^j_{s't'm'} \geq 0$;

    iii. no flows backward in time: $B^j_{stm,s't'm'} = 0$ for $t < t'$;

    iv. non-expansiveness (energy production does not exceed available resources): $\sum_{s't'm'} B^j_{s't'm',stm} < 1$.

## 3.2 Objective 2: include return flows of water and food after human consumption

In existing studies, return flows of water refer to water that is withdrawn but not consumed, subsequently re-entering the water cycle, as well as polluted flows from domestic, industrial, and agricultural sectors (Hanasaki et al., 2018; Ravar et al., 2020). This study defines return flow as the resource flow that re-emerges as a new resource source after human consumption. Illustrative examples of return flows of water include human urine and recycled water from washing and 280 cleaning. Similarly, examples of return flows of food comprise human manure and non-edible parts of food such as peels and pits. Note that return flows exclusively originate from final human consumptions.





In the water balance model, the return flows of water from human use are captured by $B^{0i}d^i$. Similarly, food also have return flows after human consumption. After integrating food, $B^{0i}d^i$ in the resource balance model can now also capture the return flows of food, thereby achieving **Objective 2.**

To clarify the transforming process of the return flows, Eq. (5) with full subscripts is presented here, shown in Eq. (5a).

$$\sum_i d^i_{hstlmc} + \sum_j e^j_{hstlmc} + \tilde{x}_{hstlmc} = A_{hstlmc,h's't'l'm'c'}\tilde{x}_{h's't'l'm'c'} + \sum_i B^{0i}_{hstlmc,h's't'l'm'c'}d^i_{h's't'l'm'c'} +$$
$$\sum_j B^j_{hstlmc,h's't'l'm'c'} e^j_{h's't'l'm'c'} + b_{hstlmc} \tag{5a}$$

In Eq. (5a), $B^{0i}_{hst\ell mc,h't'\ell'm'c'}$ represents the matrix of return coefficients for human demand $d^i_{h's't'l'm'c'}$. It quantifies the return flow of human demand type $i$, representing the outflow $(h, s, t, l, m, c)$ per unit of inflow $(h', s', t', l', m', c')$. The

term $B^{0i}_{hstlmc,h's't'l'm'c'}d^i_{h's't'l'm'c'}$ measures the return flow of human demand $i$.

Taking the generation of human urine as an example, in the resource balance equation for $(h, s, t, l, m, c)$, $B^{0i}_{hstlmc,h's't'l'm'c'}$ converts clean drinking water $d^i_{h's't'l'm'c'}$ of quality $h'$ into urine of water quality $h$ after processing in the human body. Note that $d^i_{h's't'l'm'c'}$ is derived from a different resource balance equation for $(h', s', t', l', m', c')$. Similarly, human manure resulting from food consumption, representing the return flow of human food demand, is also captured by

$B^{0i}_{hstlmc,h's't'l'm'c'}$.

For energy commodities (especially fossil fuels), their consumption typically results in the emission of greenhouse gases, which exit the WEF nexus and do not have direct impacts on the nexus. Therefore, it is assumed that there is no return flow of energy commodities after human consumption. As a result, the term $\sum_i B^{0i}d^i$ in Eq. (5) is excluded for all energy commodities. Nonetheless, this model retains the capacity to account for carbon emissions, as discussed in Appendix B.

**3.3 Objective 3: achieve transferability across spatial and temporal scales, as well as geographic contexts**

Similar to the JRB model, the WEF nexus model defines a cell as the minimum unit, through the combination of six dimensions: $h$, $s$, $t$, $l$, $m$, and $c$. Among them, the location dimension $s$ ($s = 1, ..., S$) partitions the research area into $S$ adjacent polygons, ensuring complete spatial coverage with no residual areas. There are no restrictions on the area of each polygon. By adjusting the size of spatial area represented by each $s$, the model can be applied across various spatial

resolutions, encompassing both gridded and administrative regions. The time dimension $t$ ($t = 1, ...., T$) divides the study period into $T$ time steps, covering the hydrological cycle. By varying the length of each time step $t$, the model can also adapt to different temporal resolutions, from seconds to years. These flexible properties allow this model to be transferable across different spatial and temporal scales, as well as geographic contexts, contributing to the fulfilment of **Objective 3.**





## 4 Underlying resource interactions in the resource balance model

Building upon the water balance model from the JRB model, the resource balance model for the WEF nexus has been established. While the JRB model does not explicitly introduce the concept of resource interactions, the modified resource balance model has the capability to identify and quantify all underlying resource interactions within the WEF nexus, thus fulfilling **Objective 4**.

Resource interactions occur along the resource flows, which are driven by firm activities, natural processes (e.g., water

evaporation and percolation), or human demands (e.g., generation of urine and manure). Firm activities, in particular, transform resource inputs into forms that are utilized, processed, polluted, or transported, with resource interactions taking place throughout these processes.

Based on the identification strategy, resource interactions can be classified into: In-balance resource interactions, which can be identified within a single resource balance equation for a specific combination of dimensions (i.e., time, location, soil

layer, quality, commodities, and water temperature). These interactions evolve alongside human demands, resource uses in firm activities, and natural outflows; Cross-balance resource interactions, which must be identified across resource balance equations, emerge in conjunction with resource flows spanning various dimensions.

The identification of in-balance resource interactions is driven by the equal sign of the resource balance equation. In most optimization models, resource constraints are represented as inequalities, where resource use is less than or equal to its origin

(Yue et al., 2022), representing the possibility of "free disposal" of surpluses. However, this does not hold in this model. For water, it must remain within the hydrological cycle and cannot be freely disposed of by nature. For food, aligning with the principles of the circular economy, this model also does not assume "free disposal", instead incorporating reused and recycled resources. Consequently, an equality constraint is adopted in the resource balance equation, which requires that all uses and origins of resources be explicitly accounted for, thereby fully capturing underlying in-balance resource interactions.

These in-balance resource interactions are exemplified by human demands, resource uses in firm activities, and natural outflows, which are respectively captured by $d^i$, $e^j$, and $\tilde{x}$ in Eq. (5).




The identification of cross-balance resource interactions relies on the interconnectedness of the extensive resource balance equations across six dimensions. Adhering to the principles of resource circularity, no resources will disappear; instead, they are either relocated to different locations or soil layers or transform into different qualities (and water temperature if applicable) or forms over time. This is represented by resource flows across the $H * S * T * L * M * C$ resource balance equations. The extensive resource balance equations across six dimensions, along with their interconnectedness, enables the identification of all underlying cross-balance interactions arising from natural inflows, human demands, and firm activities. These interactions are captured by $A$, $B^{0i}$, and $B^j$ respectively in Eq. (5).

### 4.1  Mutual interactions between water and energy

To ground the model in a real-world context, this subsection demonstrates how the resource balance model identifies and quantifies these interactions, using mutual interactions between water and energy as an example. Further details on other interactions are provided in Appendix C.

In the resource balance model, when $j$ represents the thermal power plant that uses cooling water for electricity generation, the plant extracts cooling water $m'$ from the river, characterized by higher quality $h'$ and lower temperature $c'$. The cooling water used by the plant is denoted as $e^j_{h'stlm'c'}$, on the left side of the resource balance equation for $(h', s, t, l, m', c')$. After usage, the water exits the firm with reduced quality $h$ and elevated temperature $c$. Through the transformation matrix $B^j_{hstlm'c,h'stlm'c'}$, the transformed water is represented as $B^j_{hstlm'c,h'stlm'c'}e^j_{h'stlm'c'}$, appearing as a new origin on the right side of the resource balance equation for $(h, s, t, l, m', c)$. The amount of cooling water consumed and the resulting temperature change depend on the specific technology employed (e.g., once-through cooling or closed-loop cooling) (Fricko et al., 2016), captured by the technology-specific matrix $B^j$.

In line with the principles of circular economy and environmental protection, wastewater generated from its use as cooling water should be treated and reused. When $j$ represents the wastewater treatment plant, the wastewater used for treatment is represented as $e^j_{hstlm'c}$. Through the corresponding transformation matrix $B^j$, the wastewater is transformed into water with improved quality and suitable temperature. The treated water can subsequently be reused, which is reflected in the model by its inclusion as a new origin in other resource balance equations.

### 4.2  Visual illustrations of resource interactions for the BTH region, China

The BTH region comprises Beijing, Tianjin, and Hebei Province. Bordered by the Bohai Sea to the east, the BTH region is located within the Haihe River basin and forms part of the North China Plain, serving as a key urban cluster. In 2015, the central government of China proposed the Outline of the Plan for Coordinated Development of the BTH Region.

To accurately capture the local context of the BTH region, the settings of variables, parameters, and indices are informed by a three-month field visit conducted between July and September 2023, including discussions with local village officials, as well as by official reports and statistical data. Within this context, human demand for the three resource commodities $i$ is



specified as urban domestic resource use and rural domestic resource use. Firm activities $j$ include river water and groundwater pumping, water transfer (import and export), artificial ecological water replenishment, water harvesting,

seawater desalination, wastewater treatment, electricity generation (thermal, hydropower, and biomass), electricity storage, electricity trade (import and export), electricity loss, organic fertilizer production, crop farming, animal farming, crop storage, crop loss, and crop trade (import and export).

As resource interactions are primarily driven by firm activities, the firm activity layer is used as an illustrative example. For simplicity, the analysis focuses on three key resource commodities: water, electricity, and crops. The origins and uses of

water within this layer, as well as the its flows between this layer and other soil layers, are depicted in Fig. 1. Detailed descriptions and visualizations for electricity and crops are in Fig. D1 and Fig. D2, provided in Appendix D.

When the resource balance model represents water, the left side of Eq. (5) corresponds to the left side of Fig. 1, which illustrates the water uses, including: (1) human water demand $\sum_i d^i$, which consists of rural domestic water use and urban domestic water use; (2) water uses in firm activities $\sum_j e^j$, which include crop farming water use, artificial ecological water

replenishment, hydropower water use, internal water export, farm animal water use, thermal power water use, and other uses, and; (3) water outflows $\tilde{x}$, which account for leakage and outflow.

The right side of Eq. (5) corresponds to the right side of Fig. 1, showing the origins of water, including: (1) return flows from human demand $\sum_i B^{0i} d^i$, including human urine after rural and urban domestic water use, which appears as a new origin after treatment; (2) flows from firm activities $\sum_j B^j e^j$, including water flows from the river layer (i.e., river water pumping

and internal water import), water flows from the aquifer zone layer (i.e., groundwater pumping), water flows from the land surface layer (i.e., water harvesting), treated wastewater deriving from water uses (i.e., rural and urban domestic water use, and thermal power water use), and other origins; (3) natural inflows $A\tilde{x}$, which refer to the inflow from the river layer; and (4) net exogenous availability $b$, which denotes the net values of external water import, seawater import, minus losses due to evaporation and external water export.



**Figure 1.** Water origins and uses within the firm activity layer and water flows across the firm activity layer. The bold parallelogram delineates the boundary of the WEF nexus. Activities within this boundary are considered endogenous, while those outsides are exogenous. Black rectangular boxes represent firm activities and human resource use, while activities without boxes represent natural flows. Red texts indicate activities where resource quality changes. Annotations here also apply to Fig. D1 and Fig. D2.

Despite the simplified depiction, Fig. 1, Fig. D1, and Fig. D2 capture various in-balance and cross-balance interactions across the six dimensions. For example, in the water dimension of Fig. 1, in-balance interactions include urban and rural domestic water use, and farm animal water use. Cross-balance interactions occur across locations (e.g., internal imports and




exports), time (all firm activities take time, and whether they span across time depends on the chosen temporal resolution),
quality (e.g., firm activities affecting water quality, marked in red), soil layers (e.g., water flows across soil layers), and
temperatures (e.g., cooling water use in thermoelectric generation). Moreover, water flows to the crops and electricity
dimensions, inducing cross-balance interactions between commodities.

All the resource interactions depicted are captured by the resource balance model, which consists of $H * S * T * L * M * C$
resource balance equations. Each equation is a combination of six dimensions: location, time, soil layer, quality, commodity,
and water temperature. These equations are interconnected, with resource flows spanning all six dimensions. Together, Fig.
1, Fig. D1, and Fig. D2 provide a holistic representation of resource flows within the nexus.

## 5 Social welfare optimization model

### 5.1 Objective 5: integrate economic, environmental, and societal considerations

Thus far, the resource balance model has incorporated water and food quality degradation as measures of environmental
quality. In line with **Objective 5**, an ideal model for the WEF nexus should integrate economic, environmental, and societal
considerations. To achieve this, the social welfare optimization is introduced, with the resource balance model serving as the
resource constraint.

The social welfare optimization model aims to maximize overall social welfare, comprising the total consumer utility
derived from resource consumption and the valuation of end-of-stock resources.

To establish the social welfare optimization model, several key assumptions are outlined:

**Assumptions 4 (Consumer utility)**: The utility function $U_i: R_+^N \to R_+$ for consumer type $i$, where $d^i = \left(d^i_{hstlmc}\right)_{(hstlmc) \in H \times S \times T \times L \times M \times C} \in R_+^N$, $N = |H| \cdot |S| \cdot |T| \cdot |L| \cdot |M| \cdot |C|$, represents consumption for each combination of resource quality ($h$), location ($s$), time ($t$), and soil layer ($l$), commodity ($m$), and water temperature ($c$), satisfies:

   i. $U_i$ is continuous on $R_+^N$ and twice continuously differentiable on $\left\{d^i \in R_+^N \middle| d^i_{hstlmc} > 0 \text{ for all } (h, s, t, l, m, c)\right\}$.

   ii. $U_i$ is non-decreasing, i.e., $\partial U_i / \partial d^i_{hstlmc} \geq 0$ for all $d^i \in R_+^N$, with $\partial U_i / \partial d^i_{hstlmc} > 0$ when $d^i_{hstlmc} > 0$.

   iii. $U_i$ is concave on $R_+^N$, with the Hessian matrix negative semidefinite on $\left\{d^i \in R_+^N \middle| d^i_{hstlmc} > 0 \text{ for all } (h, s, t, l, m, c)\right\}$.

   iv. $U_i(0) = 0$.

**Assumptions 5 (Terminal resource valuation)**: The valuation function $V: R_+^N \to R_+$ for terminal resource stocks, where $S = (S_{hslmc})_{(hslmc) \in H \times S \times L \times M \times C} \in R_+^N$, $N = |H| \cdot |S| \cdot |L| \cdot |M| \cdot |C|$, represents consumption for each combination of resource quality ($h$), location ($s$), and soil layer ($l$), commodity ($m$), and water temperature ($c$) at time $t = T$, satisfies:

   i. $V$ is continuous on $R_+^N$ and twice continuously differentiable on $\{S \in R_+^N | S_{hslmc} > 0 \text{ for all } (h, s, l, m, c)\}$.

   ii. $V$ is non-decreasing, i.e., $\partial V / \partial S_{hslmc} \geq 0$ for all $S \in R_+^N$, with $\partial V / \partial S_{hslmc} > 0$ when $S_{hslmc} > 0$.

   iii. $V$ is concave on $R_+^N$, with the Hessian matrix negative semidefinite on $\{S \in R_+^N | S_{hslmc} > 0 \text{ for all } (h, s, l, m, c)\}$.

   iv. $V(0) = 0$.





**Assumption 6 (Social welfare function):** The social welfare function $W$ is additively separable in consumer utility $U_i$ and terminal resource valuation $V$. It is continuous, concave, and non-negative.

Given **Assumptions 4**, **5**, and **6**, the social welfare function is defined as:

$$W = \sum_i \alpha^i \, U_i(d^i) + \beta V(S_T)$$

Where:

• $U_i(d^i)$ denotes the utility of human demand type $i$ from resource consumption $d^i$.

• $V(S_T)$ denotes the terminal valuation at time $T$ of the remaining stocks of resource $S_T$.

• $\beta$ is the time-discount factor, satisfying $0 < \beta \leq 1$. A lower $\beta$ implies a stronger preference for present welfare over future welfare, whereas $\beta = 1$ signifies intergenerational equity, valuing future utility equally to present utility.

• $\alpha^i$ denotes the welfare weight of human demand type $i$, such that $\sum_{i=1}^{I} \alpha^i = 1$.

• The remaining variables and parameters retain the same meanings as defined in the previous sections.

To allow for variable user fractions, the following assumption regarding their continuity is introduced:

**Assumption 7 (User fraction continuity):** The functions $\Delta(\delta^i)$, $H(\eta^j)$, $R(\rho)$ are continuous, defined on compact simplices $S^I, S^J, S$.

Building on **Assumptions 4**, **5, 6** and **7**, along with **Assumptions 1**, **2**, and **3** concerning water, food, and energy, a

constrained social welfare optimization program with variable user fractions is formalized, denoted as (P1). This program aims to allocate resources optimally among users to maximize overall consumer welfare while ensuring adherence to resource balance constraints, as outlined:

$$\max_{x, d^i, e^j, \tilde{x}, \delta^i, \eta^j, \rho} \sum_i \alpha^i \, U_i(d^i) + \beta V(S_T)$$

subject to

$$\sum_i d^i + \sum_j e^j + \tilde{x} = A\tilde{x} + \sum_i B^i d^i + \sum_j B^j e^j + b$$

$$S_T = \sum_i d^i_T + \sum_j e^j_T + \tilde{x}_T$$

$$d^i = \Delta(\delta^i)x$$

$$e^j = H(\eta^j)x$$

$$\tilde{x} = R(\rho)x$$

$$\sum_i \Delta(\delta^i) + \sum_j H(\eta^j) + R(\rho) = \iota$$

The non-convexity of the three user fraction constraints leads to multiple local optima in the optimization model, necessitating a two-level procedure to enhance convergence and computational efficiency.





- The inner problem optimizes the resource stock $x$ and the three uses of the stock (i.e., $d^i$, $e^j$, and $\tilde{x}$) while holding the corresponding user fractions (i.e., $\Delta^i$, $H^j$, and $R$) fixed. It ensures the convexity of the constraints and the existence of

optimal solutions, including the Lagrange multipliers.

- The outer problem updates user fractions based on the corresponding Lagrange multipliers from the inner problem, adjusting them along the gradient direction while ensuring they sum to one through simplex projection.

To solve the inner problem, an intermediate optimization program with fixed user fractions is formulated, denoted as (P2), as delineated:

$\max_{x, d^i, e^j, \tilde{x}, \delta^i, \eta^j, \rho} \sum_i \alpha^i U_i(d^i) + \beta V(S_T)$

subject to

$$\sum_i d^i + \sum_j e^j + \tilde{x} = A\tilde{x} + \sum_i B^i d^i + \sum_j B^j e^j + b$$

$$S_T = \sum_i d_T^i + \sum_j e_T^j + \tilde{x}_T$$

$$d^i = \Delta^i x$$

$e^j = H^j x$

$$\tilde{x} = Rx$$

$$\delta^i = \bar{\delta}^i$$

$$\eta^j = \bar{\eta}^j$$

$$\rho = \bar{\rho}$$

$\sum_i \Delta^i + \sum_j H^j + R = \iota$

By solving the optimization program, the optimal values for $x$, $d^i$, $e^j$, $\tilde{x}$, and user fractions $\delta^i$, $\eta^j$, and $\rho$, as well as Lagrange multipliers associated with each constraints, are obtained. The Lagrange multipliers corresponding to three user fractions provide crucial insights, as they represent the shadow prices, which indicate the marginal contribution to social welfare of increasing one unit of each use. Since the constraints in the optimization program are all strictly binding, they yield non-zero

shadow prices, which can be either positive or negative. These user fractions are iteratively updated based on the corresponding shadow prices to optimize resource allocation. Under **Assumptions 1-7**, there exist user fractions $\delta^*$, $\eta^*$, and $\rho^*$, such that the resulting resource allocations is a (local) welfare optimum. The proof is provided in Appendix E.

The optimization model aims to enhance economic efficiency and overall social welfare through optimal resource allocation, reflecting economic considerations of the WEF nexus model. Societal considerations are incorporated through: (1) $\beta$, the

time-discount factor, which reflects intergenerational equity by valuing the utility derived from remaining resources for future generations, and (2) $\alpha^i$, the welfare weight assigned to different consumers (e.g., urban vs. rural, female vs. male,



high-income vs. low-income), which reflects societal preferences for equitable resource distribution among the population. Thus, this WEF nexus model integrates economic, environmental, and societal considerations within the WEF nexus, achieving **Objective 5**.

## 5.2 Objective 6: ensure theoretical rigor and empirical solvability

The theoretical rigor of the social welfare optimization model is established by proving the existence of optimal user fractions for (local) welfare optimum, as outlined in the supplementary document. Empirical application can further test if this is a global welfare optimum.

Theoretical rigor being a prerequisite, empirical solvability of the model is further ensured by showing that it remains computable within a practical time frame on standard computers, given appropriate data resolution. Specifically, the model's solvability depends on the $H * S * T * L * M * C$ resource balance equations, which, though complex, remain solvable for two reasons: First, based on empirical needs, the sets of $(h, s, t, l, m, c)$, as well as $i$ and $j$, can be simplified. Second, the model's logical structure ensures matrix sparsity through (1) the absence of backward flows in time, (2) dimension-specific applicability, and (3) water flow adjacency, with flows only to adjacent locations $s$, times $t$, and soil layers $l$. This model is well-suited for implementation in GAMS, which efficiently handles sparse matrices.

## 6 Conclusion

This study presents the first transparent and comprehensive WEF nexus model that simultaneously achieves the six objectives defining an ideal model: ensuring resource security; promoting resource circularity; enabling transferability across spatial and temporal scales and geographic scopes; facilitating comprehensive identification and quantification of resource interactions; integrating economic, environmental, and societal considerations; and ensuring theoretical rigor and empirical solvability. The model's applicability is illustrated through an example of the Beijing-Tianjin-Hebei region in China. The proposed model is well-suited for simulating diverse contextual settings and testing a broader range of policy interventions. It can replicate many scenarios commonly studied in WEF nexus literature (Damerau et al., 2016; Doelman et al., 2022; Yang et al., 2016). Moreover, it also enables the exploration of novel scenarios, as illustrated in Appendix F.

The novelty of this work lies in three main contributions. First, it advances hydrological assessment by integrating water quantity and quality, including both soluble and insoluble pollutants as well as water temperature, providing a comprehensive representation of hydrological system dynamics. Second, it explicitly models water-energy interactions, demonstrating how thermoelectric generation using cooling water modifies thermal regimes and subsequently influences hydrological processes and water availability. Third, it incorporates human-water interactions by accounting for return flows of water after human consumption, emphasizing water circularity.

This work offers a novel systems model to analyse coupled human–natural systems and guide sustainable, efficient, and equitable resource management. Nonetheless, several limitations persist. Theoretically, although the linear assumption of





resource flows simplifies the model and ensures mathematical tractability, it may compromise the accuracy of actual resource flows. If the dimensions of time and location are sufficiently fine-grained, for example, accurate to the level of
minutes and centimetres, the approximation of an infinite linear representation would approach the nonlinear actual resource flows. However, achieving such precision in real-world data is challenging. This limitation is already addressed in empirical applications by applying perturbations to the system and analysing its response (Loh, 2025; Ronchetti et al., 1997). Empirically, although this model is highly adaptable to a wide range of applications, its implementation would require substantial data and personnel resources. The importance of empirical validation and numerical simulations is recognized,
and this limitation is expected to be addressed in future research.

**Appendix A: Soil layers and their interactions in the JRB model**

In the JRB model, the soil layers are classified into:

- **Land surface layer (layer 1a)**. This layer is characterized by inflows from outside the research region (e.g., rainfall, water transfer), and outflows leaving the region (e.g., immediate evaporation, water transfer). Within the system, water
flows include runoff to rivers, percolation to the root zone, and transfer of water to the anthropogenic activities (e.g., water harvesting).

- **River layer (layer 1a)**. This layer is characterized by inflows from upstream river segments and rain and outflows to downstream river segments, with water exiting rivers through evaporation. Within this layer, the runoff from the land surface layer flows into the rivers, and water can flow out to anthropogenic activities.

- **Anthropogenic activity layer (layer 1b)**. This layer encompasses all human-induced alterations to water flows, such as dams, irrigation water pumping from groundwater, wastewater treatment, drinking water treatment, return flows from consumption and water transfer (import and export).

- **Root zone layer (layer 2)**. This layer receives water from anthropogenic activity layer through irrigation and leakages, while percolation from the surface layer also adds to the water availability in this layer. Water can leave the system by
evaporation and crop evapotranspiration, as well as through leakages and percolation into the aquifer layer.

- **Aquifer zone layer (layer 3)**. This layer receives water from percolation and leakages from the root zone. Groundwater pumping transfers water from the aquifer zone to the anthropogenic activity layer, while natural springs transfers water to the land surface layer. In addition, lateral flows within aquifers are possible, although these typically will be very limited.

These soil layers collectively capture the complex interplay between natural hydrological processes and human activities,
thereby facilitating a comprehensive understanding of water dynamics. Fig. A1. presents the schematic overview of water flows within the JRB basin, including water flows across five soil layers as well as water flows entering the basin as whole (rainfall and lateral flows from outside the basin) and leaving the basin (evaporation, lateral flows leaving the basin).





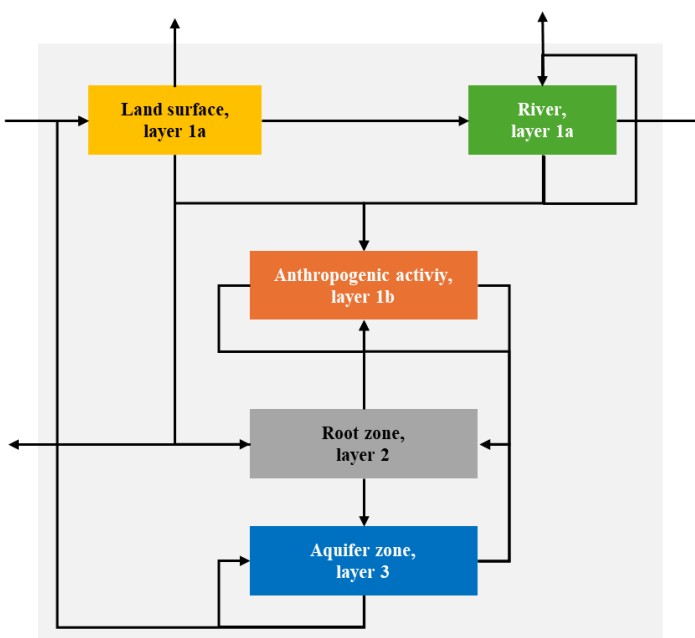

**Figure A1**. Schematic overview of water flows within the JRB basin. This figure was redrawn based on the original
illustration in "Optimal calibration and control in a regional water economy model for the Jordan River Basin" by M.A.
Keyzer (2015).

**Appendix B: Extension of the model to include carbon emissions**

As discussed in Subsection 3.2, the baseline WEF nexus model treats carbon emissions as external to the nexus and does not
directly account for them. However, the model's flexible structure allows for the integration of carbon emissions through
two distinct approaches, enabling a more comprehensive analysis of environmental impacts of the WEF nexus.

The first approach is a post-analysis calculation of carbon emissions based on energy consumption associated with key
carbon-emitting activities, including human energy use (e.g., household electricity and heating), industrial processes (e.g.,
food processing), and agricultural activities (e.g., livestock farming). Emissions are estimated by multiplying the energy
consumption of each activity by the carbon dioxide emission factor (denoted as $C_m$) for the energy commodity used (e.g.,
natural gas: 202 kg $CO_2$/MWh, diesel oil: 267 kg $CO_2$/MWh) (Willaarts et al., 2020).

The second approach integrates carbon emissions directly into the optimization model by introducing emission constraints.
Total emissions are constrained by an upper limit $\overline{Carbon}$, ensuring compliance with climate goals: $\sum_t \sum_s \sum_m (\sum_i d_{stm}^i +$
$\sum_j e_{stm}^j) C_m \leq \overline{Carbon}$. The threshold $\overline{Carbon}$ can be set according to policy guidelines, such as the EU's Fit for 55
package (targeting a 55% GHG reduction by 2030 from 1990 levels) or the U.S. Inflation Reduction Act (aiming for a 40%
cut by 2030 from 2005 levels).



The model also accounts for the energy mix through the energy production, $B^j e^j$, which captures contributions from multiple sources, including fossil fuels, hydropower, solar energy, and wind power. Policy-driven targets—such as the EU's goal of 42.5% renewable energy by 2030 (Specifically, Germany's coal phase-out by 2038) or projected a 50% reduction in coal-based electricity by 2030—can inform constraints on energy mix in the model.

**Appendix C**: **Resource interactions within the WEF nexus**

This appendix elaborates on each resource interaction within the WEF nexus, providing real-world examples and demonstrating how the resource balance model can identify and quantify these interactions. Resource interactions within the WEF nexus are illustrated in Fig. C1.

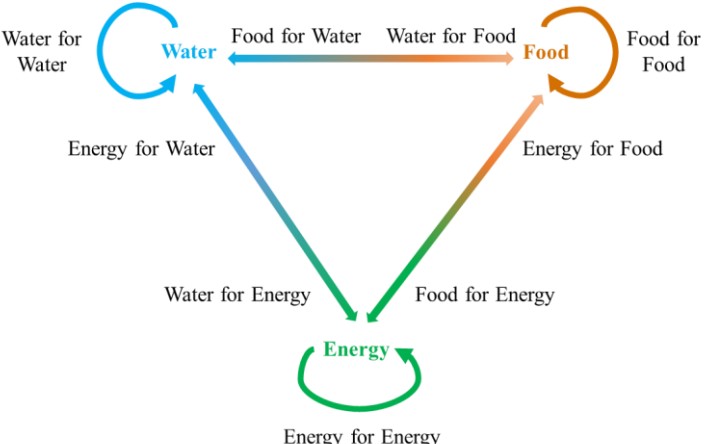

570                                     **Figure C1**. Resources interactions within the WEF nexus.

**Water for food**

In the WEF nexus, the water for food interactions consist of:

- Water inputs into the food system, such as water use for irrigation, water use for food processing, and water use for farm animals. These in-balance interactions, exemplified by resource uses in firm activities, are identified by $e^j$ in Eq. (5). For
instance, given specific water quality $h$, water temperature $c$, location $s$, and time $t$, when commodity $m'$ represents water and soil layer $l$ denotes the firm activity layer, the resource balance equation can be formulated for $(h, s, t, l, m', c)$. In this equation, for firm activity $j$ representing agricultural irrigation, the irrigation water use is identified and quantified by $e^j_{hstlm'c}$ on the left side of Eq. (5). This term reflects the water footprint of irrigation, encompassing both blue water from irrigation and green water from rainfall.

- Food-related impacts caused by the water system, such as crop yields benefiting from irrigation and changes in food quality due to irrigation water quality. These impacts represent the consequent cross-balance resource interactions following resource uses in firm activities, captured by $B^j$. Continuing with the previous example, after crops absorb





irrigation water, they grow, mature, and eventually transform into usable crops. The resource balance model reflects this process as resource flows across the commodity dimension, from water $m'$ to crop $m$. Specifically, the irrigation water

$e^j_{hstlm'c}$ on the left side of the resource balance equation for $(h, s, t, l, m', c)$ flows to the resource balance equation for $(h, s, t, l, m, c)$. By the transformation matrix $B^j_{hstlmc,hstlm'c}$, the term $B^j_{hstlmc,hstlm'c} e^j_{hstlm'c}$ represents the crop yield resulting from the firm activity of agricultural irrigation. This transformed flow then appears as a new origin on the right side of the resource balance equation for crop $m$.

**Water for energy**

The water for energy interactions consist of:

- Water inputs into the energy system, such as water used for hydropower generation and pumped-storage hydroelectricity, cooling water used for fuel processing and thermoelectric generation, and water use for mining and coal washing. These in-balance interactions, exemplified by resource uses in firm activities, are identified by $e^j$ in Eq. (5).

- Energy-related impacts caused by the water system, such as generated hydropower and generated electricity from
pumped-storage, and energy footprint associated with water supply and wastewater treatment (Gu et al., 2019). These are cross-balance resource interactions captured by $B^j$. Similar to agricultural irrigation, the water used for hydropower generation, denoted as $e^j$ on the left side of the resource balance equation, flows to the right side of another resource balance equation for electricity as a new origin. Unlike irrigation, hydropower generation does not consume water; instead, it uses the gravitational potential energy of water. After electricity generation, the water eventually returns to the
river. The firm activity-specific $B^j$ captures this distinction, which represents the technical details of each firm activity, including the quantities of resources consumed and changes in resource quality.

**Energy for water**

The energy for water interactions consist of:

- Energy inputs into the water system, such as electricity used for groundwater pumping, energy used for seawater
desalination, and energy used for drinking water treatment and wastewater treatment. These in-balance interactions are captured by $e^j$.

- Water-related impacts caused by the energy system, such as polluted water from coal washing, elevated temperature of discharged cooling water, and reclaimed water after treatment. These cross-balance resource interactions are captured by $B^j$.

**Energy for food**

The energy for food interactions consist of:

- Energy inputs into the food system, such as electricity used for irrigation, diesel used for operating agricultural machines, chemical fertilizer application, energy used for food processing, transportation and supply, and food storage. These are in-balance interactions, captured by $e^j$. For instance, when $j$ represents crop trade facilitated by diesel-powered
transportation, diesel $m$ is used to transport crop $m'$ of human consumption quality $h'$ from location $s'$ at time $t'$. The



diesel used for transportation is denoted as $e^j_{s't'lm}$, appearing on the left-hand side of the resource balance equation for $(s', t', l, m)$.

- Food-related impacts caused by the energy system, such as food loss during trade and quality degradation during storage. Food storage and trade are prevalent strategies in practice used to address the spatial and temporal disconnects between demand and supply (Scanlon et al., 2017). As explained before, all the accompanying transformations in location, time, and quality dimensions can be captured by $B^j$.

**Food for water**

- The food for water interactions represent the water-related impacts caused by the food system, such as polluted water due to excess fertilizer and pesticide use in agriculture. These are cross-balance interactions, captured by $B^j$. A prominent example of such an interaction is virtual water embedded in food trade, which captures the indirect transfer of water resources through traded agricultural goods (Salmoral and Yan, 2018). This virtual water is computed as the product of crop-specific irrigation water footprint (as detailed in the water for food subsection) and the quantity of traded crops. To illustrate, consider the trade of corn, denoted as firm activity $j$. Let $e^j_{hs't'lm}$ represent the quantity of corn $m$ of quality $h$, exported from location $s'$ at time $t'$. This exported amount is transformed through the matrix $B^j_{hstlm', hs't'lm'}$, which maps exports to imports. Accordingly, the quantity of corn imported by location $s$ at time $t$ is given by $B^j_{hstlm, hs't'lm} e^j_{hs't'lm}$.

**Food for energy**

The food for energy interactions consist of:

- Food inputs into the energy system, such as agricultural byproducts (e.g., crop straw) used for electricity generation, human and animal manure used for biogas generation and organic fertilizer production (Yue et al., 2022), and energy crops for biofuel production. These in-balance interactions are captured by $e^j$.
- Energy-related impacts caused by the food system, such as energy footprint of agricultural irrigation (Lu et al., 2021). Specifically, the calculation of this footprint involves two key components: (1) the energy used to extract and pump water from aquifers or surface water bodies, as discussed in the energy for water interaction; and (2) the energy used during irrigation processes in agricultural fields, as outlined in the energy for food interaction.

**Water for water**



The water for water interactions refer to the water inputs into the water system, and the water-related impacts caused by the water system. As water is the sole commodity under its class, the water for water interactions primarily concern the changes of water in quality, water temperature, time, location, and soil layer. Explicit examples for this interaction include the production of clean water (e.g., wastewater treatment, drinking water treatment, and seawater desalination), the production of

wastewater (e.g., industrial and agricultural water pollution, human and animal urine from water drinking, and discharged cooling water from power plants), water transfer, and groundwater pumping.

An example where water flows across soil layers is presented here. When $j$ represents groundwater pumping, which extracts water from the aquifer layer $l'$ for use in firm activity layer $l$, the pumped groundwater $m'$, with quality $h'$ at location $s$ and time $t$ in aquifer layer $l'$ at temperature $c'$, is denoted as $e^j_{h'stl'm'c'}$ on the left side of the resource balance equation for

$(h', s, t, l', m', c')$. The pumped groundwater then becomes a new origin in the firm activity layer $l$, realized through the transformation matrix $B^j_{h'stlm'c',h'stl'm'c'}$.

The water for water interactions are often interdependent with energy and food flows, as outlined in the water for food, water for energy, energy for water, and food for water interactions. However, there are exceptions where water for water interactions occur independently: (1) natural water flows, including quality transformations (e.g., percolation and

evaporation) across the dimension $h$, and water retention, which involves transformations solely across the time dimension $t$. Both are governed by the matrix of natural water flow coefficients $A$. (2) human water demand, which causes the transformation of clean drinking water into urine within the human body. As specified in Subsection 3.2, urine is treated as a return flow following human consumption, with this interaction captured by $B^{0i}$.

**Food for food**

The food for food interactions refer to the food inputs into the food system, and the food-related impacts caused by the food system. Considering various commodities under food class, this interaction can be exemplified by the transformation of food across location, time, quality and form. Illustrative examples for this consist of food processing, food spoilage, feed for animal farming, and the return of crop stalks to the field.

Building upon the example of crop trade via diesel-powered transportation, as discussed in the energy for food interactions,

crop $m$ is transported from location $s'$ to location $s$ over the period from $t'$ to $t$, during which the crop deteriorates from human consumption quality $h'$ to animal feed quality $h$. The exported crop is denoted as $e^j_{h's't'lm'}$, on the left side of the resource balance equation for $(h', s', t', l, m')$. Using the transformation matrix $B^j_{hstlm',h's't'lm'}$, the exported crop is converted into the imported crop of quality $h$ at location $s$ at a later time $t$, represented as $B^j_{hstlm',h's't'lm'}e^j_{h's't'lm'}$ on the right side of the resource balance equation for $(h, s, t, l, m')$.

This example illustrates the interdependence of resource flows, as further exemplified in water for food, food for water, energy for food, and food for energy interactions. Similar to water for water, food for food interactions are often closely linked with energy or water flows, occurring simultaneously and influencing one another. An exception where food for food



interactions occur independently is human food demand, which leads to the conversion of food into manure within the body. Human manure is considered the return flow of food, and this transformation is captured by $B^{0i}$.

**Energy for energy**

The energy for energy interactions refer to the energy inputs into the energy system, and the energy-related impacts caused by the energy system. Given the homogeneous quality of energy, the energy for energy interactions are exemplified by changes in energy quantities across time (e.g., electricity storage) and across location (e.g., natural gas transportation), and the transformation between energy commodities (e.g., conversion of electricity from coal, natural gas, and crude oil) (Ravar

et al., 2020).

When $j$ represents the coal plant that burns coal $m'$ to generate electricity $m$, the coal utilized by the plant is denoted as $e^j_{stlm'}$. Through the transformation matrix $B^j_{stlm,stlm'}$, the coal is transformed into electricity, represented by $B^j_{stlm,stlm'} e^j_{stlm'}$, which then appears as a new origin in the resource balance equation for $(s,t,l,m)$. On the right side of the equation, $b$ captures net exogenous availability, encompassing electricity generation from nuclear power.

The energy for energy interactions are often interdependent with water and food flows, as outlined in the water for energy, energy for water, energy for food, and food for energy interactions. An exception is human energy demand, which represents the direct consumption of energy by humans.

**Appendix D**: **Visual illustrations of electricity and crops for the BTH region, China**

When the resource balance model represents the electricity, the left side of Eq. (5) corresponds to the left side of Fig. D1,

which illustrates electricity uses, including: (1) human electricity demand $\sum_i d^i$, which consists of rural and urban domestic electricity use; (2) electricity uses in firm activities $\sum_j e^j$, which include electricity for water pumping, electricity for seawater desalination, electricity for wastewater treatment, electricity for crop farming, electricity for animal farming, electricity for crop storage, electricity loss, internal electricity export, electricity storage, and other uses. As discussed in Section 3, energy does not have natural flows, so the resource balance model for electricity excludes nature outflows $\tilde{x}$.

The right side of Eq. (5) for electricity corresponds to the right side of Fig. D1, which shows the origins of electricity, including: (1) flows from firm activities $\sum_j B^j e^j$, including released electricity storage, internal electricity import, electricity from biomass, hydropower, thermal power, wind, solar power, and other origins; (2) net exogenous availability $b$, which denotes the net values of external electricity import minus the externally exported electricity. As energy does not have natural flows, there are no natural inflows $A\tilde{x}$ in the resource balance model for electricity. Additionally, as discussed in

Subsection 3.2, electricity has no return flows from human demand $\sum_i B^{0i} d^i$.



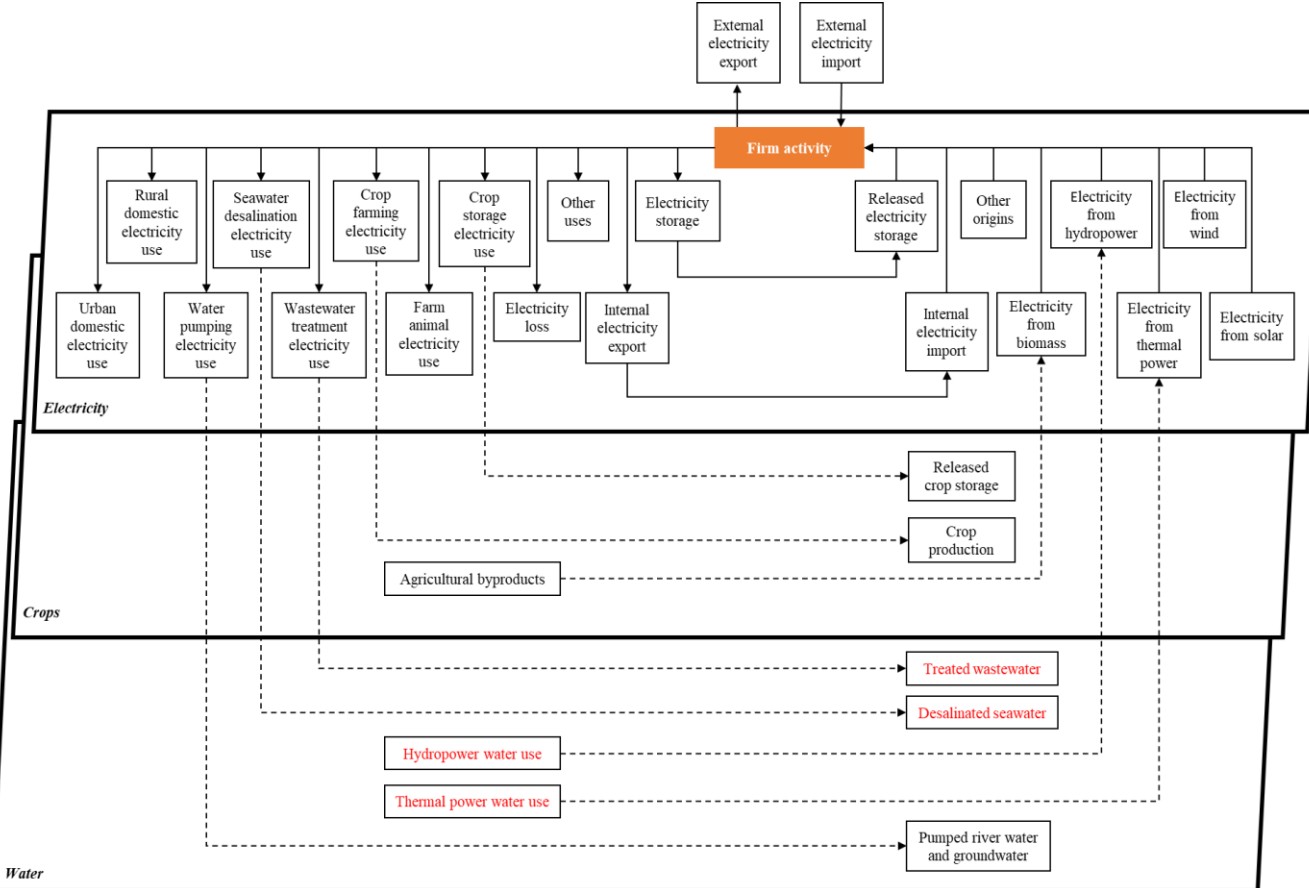

**Figure D1.** Electricity origins and uses within the firm activity layer and water flows across the firm activity layer.

When the resource balance model represents crops, the left side of Eq. (5) corresponds to the left side of Fig. D2, which illustrates crop uses, including: (1) human crop demand $\sum_i d^i$, which consists of rural and urban domestic crop use; (2) crop uses in firm activities $\sum_j e^j$, which include agricultural byproducts for biomass power generation, energy crop for biofuel generation, farm animal feed use, crop loss, crop storage, internal crop export, and other uses. As discussed in Section 3, crops do not have natural flows, so the resource balance model for crops excludes nature outflows $\tilde{x}$.

The right side of Eq. (5) for crops corresponds to the right side of Fig. D2, which shows the origins of crops, including: (1) return flows from human demand $\sum_i B^{0i} d^i$, including human manure after rural and urban domestic crop use, which serves as organic fertilizer after treatment; (2) flows from firm activities $\sum_j B^j e^j$, including internal crop import, released crop storage, crop production, organic fertilizer production from animal manure, and other origins; (3) net exogenous availability $b$, which denotes the net values of external crop import minus the externally exported crop. As crop does not have natural flows, there are no natural inflows $A\tilde{x}$.



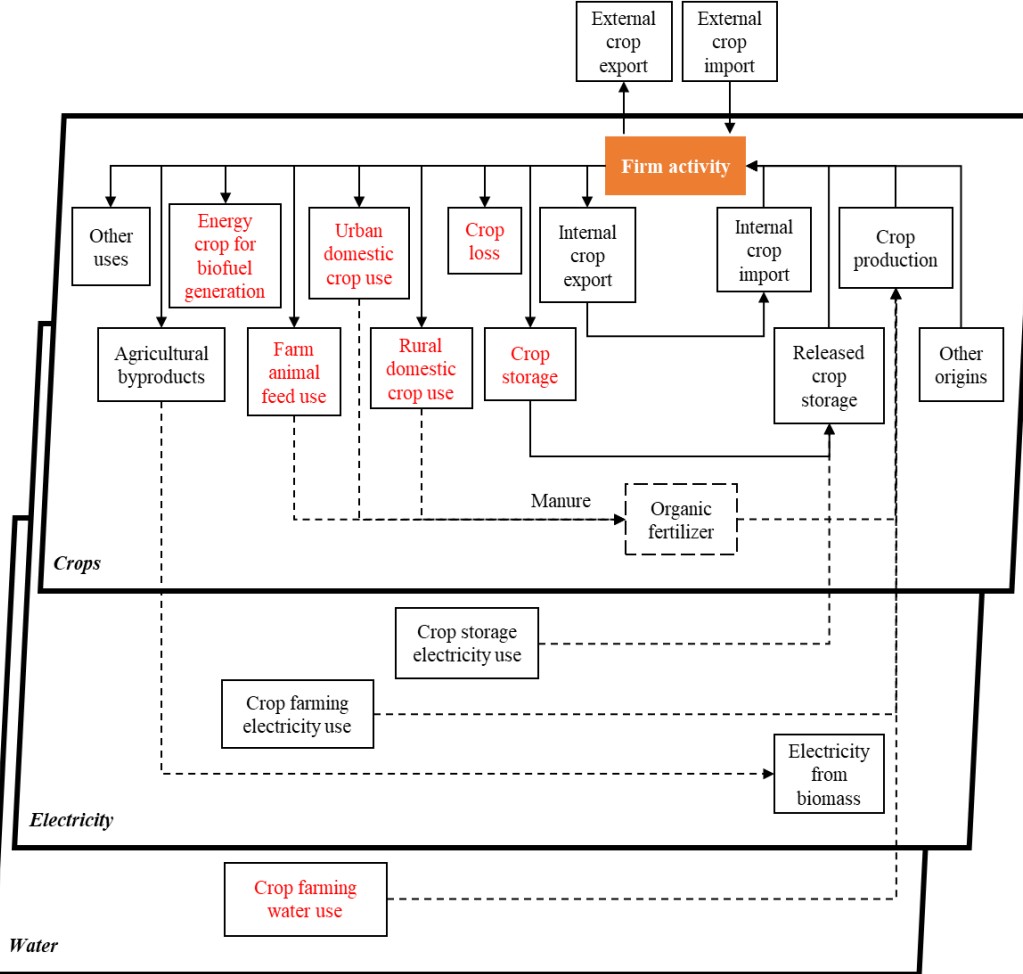

**Figure D2.** Crop origins and uses within the firm activity layer and water flows across the firm activity layer.

Since Fig. 1, Fig. D1, and Fig. D2 focus on the three commodities, not all cross-commodity interactions are depicted. For example, in Fig. D2, the interactions between crop and commodities other than electricity and water are not fully illustrated. Energy crops used for biofuel generation are converted into biofuel, which then enters the resource balance equation for biofuel as an origin. Similarly, crops used for feeding farm animals contribute to farm animal production and are accounted

for in the resource balance equation for farm animals as an origin. While these interactions are captured by the resource balance model, they are not explicitly shown here, as the focus remains on crop-related interactions within the firm activity layer and its interactions with electricity and water.

**Appendix E: Proof of existence of optimal user fractions for (local) welfare optimum**





This appendix provides a formal proof that, under **Assumptions 1-7**, there exist user fractions $\delta^*$, $\eta^*$, and $\rho^*$ in the social
welfare optimization problem with variable user fractions such that the resulting resource allocation is a (local) welfare optimum.

A two-level iterative procedure is proposed to solve the non-convex optimization problem (P1). The procedure decomposes (P1) into an inner convex problem (P2) and an outer problem that updates user fractions. The proof proceeds in two steps: (1) construct a fixed-point mapping and prove a fixed point exists; (2) show the fixed point yields a (local) welfare optimum
(Ginsburgh and Keyzer, 1997).

**Step 1: Construction of the fixed-point correspondence**

Given the concavity of $U_i$ and $V$ from **Assumptions 4** and **5** and the linearity of constraints from **Assumptions 1–3**, the inner program (P2) is a convex optimization problem.

At the optimum, the Jacobian matrix of equality constraint gradients is assumed to have full row rank, satisfying the Linear
Independence Constraint Qualification and ensuring the existence of Lagrange multipliers for all equality constraints, including those on user fractions $\delta^i$, $\eta^j$, and $\rho$. The corresponding Lagrange multipliers $(\lambda_1^i, \lambda_2^j, \lambda_3)$ in (P2) serve as shadow prices, indicating the marginal impact on social welfare of adjusting each user fraction.

Define a correspondence $M^\circ: A \Rightarrow \bar{M}$, where $A = S^I \times S^J \times S$ is the compact set of resource user fractions and $\bar{M}$ is the convex hull of Lagrange multipliers $(\lambda_1^i, \lambda_2^j, \lambda_3)$. For given $(\bar{\delta}^i, \bar{\eta}^j, \bar{\rho})$, $M^\circ(\bar{\delta}^i, \bar{\eta}^j, \bar{\rho})$ yields the multipliers from (P2)'s
Karush-Kuhn-Tucker conditions.

By **Assumption 7**, $\Delta(\delta^i), H(\eta^j), R(\rho)$ are continuous. By Berge's Maximum Theorem (Berge and Patterson, 1997), the solution and multipliers are upper semicontinuous in $\bar{\delta}^i, \bar{\eta}^j, \bar{\rho}$. The set $\bar{M}$ is compact and convex, and $M^\circ(\bar{\delta}^i, \bar{\eta}^j, \bar{\rho}) \neq \{0\}$.

Define a continuous function $h: A \times \bar{M} \to A$, updating user fractions:

$$h(\bar{\delta}^i, \bar{\eta}^j, \bar{\rho}, \lambda_1^i, \lambda_2^j, \lambda_3) = (\delta^{i''}, \eta^{j''}, \rho'')$$

1.   Compute initial update:

$$\delta^{i'} = max(\bar{\delta}^i + \epsilon_1 \lambda_1^i, 0)$$
$$\eta^{j'} = max(\bar{\eta}^j + \epsilon_2 \lambda_2^j, 0)$$
$$\rho' = max(\bar{\rho} + \epsilon_3 \lambda_3, 0)$$

Where $\epsilon_1, \epsilon_2, \epsilon_3 > 0$ are step sizes.

2.   Compute weighted averages:

$$\delta^{i''} = \mu_1[(1 - \gamma_1)\bar{\delta}^i + \gamma_1 \delta^{i'}]$$
$$\eta^{j''} = \mu_2[(1 - \gamma_2)\bar{\eta}^j + \gamma_2 \eta^{j'}]$$
$$\rho'' = \mu_3[(1 - \gamma_3)\bar{\rho} + \gamma_3 \rho']$$

Where $\gamma_1, \gamma_2, \gamma_3 \in (0,1)$ are weights, and $\mu_1, \mu_2, \mu_3 > 0$ are scaling factors ensuring:



$$\sum_i \Delta\left(\delta^{i''}\right) + \sum_j H\left(\eta^{j''}\right) + R(\rho'') = \iota$$

Since $\delta^{i'}, \eta^{j'}, \rho'$ are non-negative and at least one is positive, such scaling is feasible.

Define a correspondence $G: C \to C$, where $C = A \times \bar{M}$: $G\left(\bar{\delta}^i, \bar{\eta}^j, \bar{\rho}, \lambda_1^i, \lambda_2^j, \lambda_3\right) = \left\{\delta^{i''}, \eta^{j''}, \rho'', \lambda_1^{i*}, \lambda_2^{j*}, \lambda_3^* \middle| \left(\lambda_1^{i*}, \lambda_2^{j*}, \lambda_3^*\right) \in M^\circ\left(\bar{\delta}^i, \bar{\eta}^j, \bar{\rho}\right), \left(\delta^{i''}, \eta^{j''}, \rho''\right) = h\left(\bar{\delta}^i, \bar{\eta}^j, \bar{\rho}, \lambda_1^{i*}, \lambda_2^{j*}, \lambda_3^*\right)\right\}$

Given that $C$ is compact and convex, $M^\circ$ is upper semicontinuous, and compact, convex valued, and $h$ is continuous, $G$ is upper semicontinuous, and compact, convex valued.

To establish the existence of a fixed point, Kakutani's fixed point theorem is employed, stated as follows:

Let $C \subset R^s$ be a nonempty compact convex set and $G$ a correspondence $C \Rightarrow C$ which is uppersemicontinuous in $C$ and which is such that for all $a \in A$, $G(a)$ is nonempty and convex. Then there exists a value $c^*$ such that $c^* \in G(c^*)$ (Kakutani, 1941).

Since these conditions hold in the model, there exists a fixed point:

$$\left(\delta^{i*}, \eta^{j*}, \rho^*, \lambda_1^{i*}, \lambda_2^{j*}, \lambda_3^*\right) \in G\left(\delta^{i*}, \eta^{j*}, \rho^*, \lambda_1^{i*}, \lambda_2^{j*}, \lambda_3^*\right)$$

Satisfying:

$$\left(\lambda_1^{i*}, \lambda_2^{j*}, \lambda_3^*\right) \in M^\circ\left(\delta^{i*}, \eta^{j*}, \rho^*\right)$$

$$\left(\delta^{i*}, \eta^{j*}, \rho^*\right) = h\left(\delta^{i*}, \eta^{j*}, \rho^*, \lambda_1^{i*}, \lambda_2^{j*}, \lambda_3^*\right)$$

**Step 2: The fixed point is a (local) welfare optimum**

The resource allocation $x^*, d^{i*}, e^{j*}, \tilde{x}^*$ at the fixed point is a (local) welfare optimum for (P1). At the fixed point:

$$d^{i*} = \Delta\left(\delta^{i*}\right)x^*$$

$$e^{j*} = H\left(\eta^{j*}\right)x^*$$

$$\tilde{x}^* = R(\rho^*)x^*$$

$$\sum_i \Delta\left(\delta^{i*}\right) + \sum_j H\left(\eta^{j*}\right) + R(\rho^*) = \iota$$

The fixed-point condition implies:

$$\delta^{i*} = max\left(\delta^{i*} + \epsilon_1\lambda_1^{i*}, 0\right) \implies \lambda_1^{i*} \approx 0 \text{ (if } \delta^{i*} > 0)$$

$$\eta^{j*} = max\left(\eta^{j*} + \epsilon_2\lambda_2^{j*}, 0\right) \implies \lambda_2^{j*} \approx 0 \text{ (if } \eta^{j*} > 0)$$

$$\rho^* = max(\rho^* + \epsilon_3\lambda_3^*, 0) \implies \lambda_3^* \approx 0 \text{ (if } \rho^* > 0)$$

By **Assumptions 4–6**, $W$ is concave in $d^i$ and $S_T$. The Hessian with respect to $x^*, d^{i*}, e^{j*}, \tilde{x}^*$ is semi-negative definite at the fixed point, ensuring a local maximum. Although $\Delta(\delta^i), H(\eta^j), R(\rho)$ may introduce non-convexity, the Karush-Kuhn-Tucker conditions and fixed-point stability imply a local optimum. Due to the non-satiation of the utility function, not all $\delta^{i*}$, $\eta^{j*}, \rho^*$ will be zero. Hence, there exist a set of non-zero $\delta^{i*}, \eta^{j*}, \rho^*$ such that $x^*, d^{i*}, e^{j*}, \tilde{x}^*$ is a local welfare optimum for





(P1). Global optimality of the equilibrium is not ensured; there may be multiply equilibria, and starting values for the user

fractions may imply convergence to a local optimum that is not the global one.

**Appendix F**: **Potential scenarios in the WEF nexus model**

Empirical applications of the WEF nexus often conduct scenario simulations to explore potential changes in resource quantity, quality, and interactions under future dynamics. The primary scenarios of interest include socio-economic change, environmental change, climate change, policy intervention, and technological advancement (European Commission. Joint

Research Centre., 2019; World Economic Forum, 2011). The proposed model can be used to simulate these scenarios, offering distinct advantages, particularly in addressing **Objectives 3, 4** and **5**:

- The model's transferability across spatial and temporal scales allows it to examine resource trades both within regions and between regions, providing valuable insights into transboundary resource management and integrated governance.
- Unlike other models, this model comprehensively captures the underlying resource interactions within the WEF nexus. As

a result, it ensures that external shocks are reflected across all interconnected resources, providing more accurate and reliable outcomes from a holistic systems perspective.

- This model integrates environmental, economic, and societal considerations, facilitating the optimization of resource allocations. By integrating these consideration, this model leverages the potential of the WEF nexus to tackle complex real-world policy challenges.

Beyond the commonly studied scenarios, this model explicitly incorporates resource circularity (**Objective 2**), a critical sustainability challenge that has rarely been simulated empirically (Egger and Keuschnigg, 2024; European Commission. Joint Research Centre., 2019). By endogenizing the return flows of water and food after human consumption, this model can simulate scenarios related to various circularity policy options. Additionally, the comprehensive water quality measurement of the model (**Objective 1**) enables the design of scenarios that account for specific pollutants (both soluble and insoluble,

such as heavy metals) and changes in water temperature. The inclusion of food quality (**Objective 1**) further allows for simulations of scenarios exploring the trade-offs between crop uses for human consumption and biofuel production.

Table F1 presents some potential scenarios in the model, detailing the necessary adjustments. Note that this table is intended to be explanatory, not exhaustive. Moreover, it is worth noting that those outlined scenarios are not isolated but can be integrated to capture the nexus's response to complex shocks (Van Vuuren et al., 2019; Zhang et al., 2024). This makes the

model particularly suitable for addressing sustainability challenges in small island developing states, where limited resources, small size, and remoteness intensify these issues (Crisman and Winters, 2023).

**Table F1.** Potential scenarios in the WEF nexus model

| Scenario ID | Scenario | Key assumption | Existing WEF nexus research | Changes in sets of variables, parameters, and indexes of the model |
|---|---|---|---|---|





| ID | Name | Description | Reference | Representation |
|----|------|-------------|-----------|----------------|
| S0 | Baseline | No policy intervention, current resource production and use | —— | No changes |
| W1 | Water scarcity | Declining surface and groundwater availability, coupled with rising water use | (Giuliani et al., 2022; Kahil et al., 2018) | $x$: surface and groundwater availability, when $l$ refers the river or aquifer layer; $d^i$: water use of human demand $i$; $e^j$: water use of firm activity $j$ |
| W2 | Water quality degradation | Increased pollutants in water | (Fan et al., 2019) | $h$: water quality, represented by mixed flows of pure water flows and pollutant flows |
| W3 | External water transfer | Variations in interregional water supply | (Wu et al., 2022; Xu et al., 2020; Zhang et al., 2024) | $b$: net exogenous water availability |
| W4 | Internal water transfer | Water reallocation across regions | (Yang et al., 2016) | $e^j$: volume of water transfer, when $j$ is water transfer |
| W5 | Water circularity | Reuse of desalinated and reclaimed water, and incorporation of return flows | None | $B^j e^j$: desalinated/reclaimed water; $B^{0i} d^i$: return flows after human use |
| E1 | Renewable energy expansion | Shift from fossil to renewable energy sources | (Damerau et al., 2016; Purwanto et al., 2021; Wu et al., 2022; Yang et al., 2016; Zhang et al., 2024; Zhong et al., 2021) | $B^j e^j$: production of specific energy commodity by firm activity $j$, such as biofuel, hydropower, wind power, and solar power |
| E2 | Increasing energy use | Higher energy use from firms and human | (Giuliani et al., 2022; Hussien et al., 2017; Purwanto et al., 2021) | $d^i$: energy use of human demand $i$; $e^j$: energy use of firm activity $j$ |
| F1 | Enhanced | Improvements in | (Giuliani et al., | $B^j e^j$: crop production, where $B^j$ indicates |



| | | irrigation efficiency and crop yields | 2022; Van Vuuren et al., 2019; Xu et al., 2020) | irrigation efficiency and agricultural productivity |
| --- | --- | --- | --- | --- |
| | agricultural productivity | | | |
| F2 | Crop switching | Shift to less water-intensive crops | (Kulat et al., 2019; Vahabzadeh et al., 2023) | $m$: crop types; $B^j e^j$: crop production |
| F3 | Food self-sufficiency | Change in local vs. imported food share | (Daher et al., 2022) | $B^j e^j$: local food; $b$: imported food |
| F4 | Food waste reduction | Less household food waste | (Van Vuuren et al., 2019) | $B^j e^j$: food waste, where $B^j$ indicates the transformation coefficient |
| F5 | Reuse of food waste | Increased utilization of food waste for biogas generation | None | $B^j e^j$: energy transformed from food waste, where $B^j$ represents the transformation coefficient |
| F6 | Cultivated meat | Increase in lab-grown meat production | None | Introduce a new firm activity $j$; $B^j e^j$: production of cultivated meat |
| C1 | Changes in precipitation | Altered precipitation levels and patterns | (Giuliani et al., 2022; Wu et al., 2022) | $b$: net exogenous water availability, including precipitation |
| C2 | Water temperature rise | Increased water temperature driven by global warming | None | $c$: water temperature |
| S1 | Population growth | Growing population, leading to increasing human resource demand | (Giuliani et al., 2022; Purwanto et al., 2021; Wu et al., 2022) | $d^i$: resource use of human demand $i$ |
| S2 | Dietary shift | More plant-based diets, less meat consumption | (Kheirinejad et al., 2024; Van Vuuren et al., 2019; Xu et al., 2020) | $d^i$: food consumption by commodity type $m$ |



| S3 | Urbanization | Shift from rural to urban population | (Zhang and Wang, 2024) | Reclassification of human demand $i$ by location $s$ |
|---|---|---|---|---|

Note: In this table, scenario IDs are categorized as follows: W- for water-related scenarios, E- for energy-related scenarios, F- for food-related scenarios, C- for climate change scenarios, and S- for socio-economic scenarios

*Code and data availability.* The study develops a theoretical model. No datasets or computer code were generated or analysed.

*Author contribution.* WL, PJW, and LvW conceptualized the study; WL and LvW developed the model; WL performed the analysis and prepared the original draft; PJW and LvW supervised the project and revised the manuscript.

*Competing interests.* The contact author has declared that none of the authors has any competing interests.

*Acknowledgements.* WL received support from the China Scholarship Council of the Ministry of Education, P. R. China (grant number 202106360024) and the Erasmus+ grant KA171 (call 2022) for funding fieldwork in China. We are grateful for the valuable feedback received during presentations at 2025 World Conference on Natural Resource Modeling, Tinbergen Institute PhD seminar, and Economics Department at Vrije Universiteit Amsterdam.

*Financial support.* This research has been supported by the China Scholarship Council of the Ministry of Education, P. R.
China (grant number 202106360024) and the Erasmus+ grant KA171 (call 2022).

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
