# Peer review of "Towards an ideal water-energy-food nexus model: moving beyond silos to integrated resource governance"

_EGUsphere, 2025_

## Author Comment (AC1)

**Author Comment**

We sincerely thank you for the time and expertise you devoted to reviewing our manuscript. We appreciate your thorough and constructive comments, which prompted us to clarify the scope and clearly demonstrate the practical applicability.

Due to the tight time constraints of the open discussion, we provide below an initial version of responses. Comments that pertain to the same underlying issue have been grouped and addressed collectively. A more detailed, point-by-point response will be provided during the formal revision stage.

**Response to Major Concerns**

1. **Comment:** The paper presents a purely theoretical WEF nexus model that is rigorous in theoretical development but lacking in applicability and demonstration. The claim that the model is 'ideal' being generally applicable is, in my opinion, overstating things substantially as there is absolutely no evidence to support the claim. This wording should be tempered considerably as the claim is not backed up. There are no results to support any such claims, nor indeed to show that the model is even functional. For this reason, I also find the claims that the 'model' achieves six objectives set out in the paper rather a stretch. In principle perhaps it does, but until there is empirical support, I feel the claims should be significantly reduced.

   **Response:** Thank you for the constructive comment regarding the applicability of the model. We fully agree that the applicability of a theoretical model cannot be asserted without empirical calibration and validation, and that such claims require clear supporting evidence. We also agree that, in the absence of empirical results, claims concerning the model's ability to achieve six objectives should be interpreted as potential or conceptual properties, rather than empirically demonstrated outcomes. These claims will therefore be carefully reconsidered and tempered.

   In a separate ongoing empirical study on cross-sectoral and cross-regional coordination in the Beijing-Tianjin-Hebei region, we are applying the theoretical model developed in this manuscript. Data collection for this application has been completed, and model calibration and validation are planned. This ongoing work demonstrates the operational feasibility of the proposed model.

   To directly address your concerns regarding applicability, we plan to revise the manuscript in the following aspects:

   1) **Tempering and reframing claims throughout the paper.** All statements implying empirical applicability will mention that the process of calibration takes quite some efforts and may require additional assumptions. We will also temper the claims on demonstrated performance.
   2) **Clarifying the scope and limitations of the current study.** We will explicitly state that the present paper focuses on model construction and theoretical properties and highlight better how these properties open a clear way for empirical application, but we will not claim empirical validation yet.
   3) **Adding a clear roadmap for empirical applications of the theoretical model.** As preliminarily illustrated below, an explicit roadmap will be added to outline the steps required for empirical implementation.

   We believe these revisions will substantially improve the clarity and credibility of the discussion on applicability and better align the claims with what is demonstrated in the current study.

[Figure]

Fig.1. Roadmap for practical implementation of the proposed theoretical model. This study focuses on the development of the theoretical model and, while it does not present the full implementation roadmap, it provides key contributions to several phases: the theoretical model is fully developed in Phase 0;

Phases 1, 2, and 3 are illustrated using the BTH region as an example; and the scenario design in Phase 5 can refer to Table F1. The remaining phases represent future steps beyond the scope of this study.

**2. Comment:**

- I also feel that there is no such thing as an 'ideal' nexus model that is generally applicable across spatial and temporal scales. Over 15 years of nexus research have taught me that for any nexus model to have any practical value beyond an academic exercise, the model must be developed contextually, and often in collaboration with local experts and stakeholders. For this reason, I again suggest that the authors temper their claims of the applicability, generalisability, and indeed usefulness of the purely theoretical mode developed in the paper. I would also suggest that the authors consider this a bit more deeply rather than overstating the ability of a model that as yet has no results to support it.

- Line 45-46: the note on a model having to be transferable. Please see my general comment above.

- While I appreciate the theoretical development, the model runs the real risk of being just that – a theoretical construct with little to no practical, real-world relevance. This would only add to criticisms of the nexus.

- The claim that the model was illustrated through the example of the Beijing-Tianjin-Hebei region should be removed. There is no evidence to support this claim. It is a theoretical construct only.

**Response:** Thank you for the insightful comment regarding the generalisability, transferability, and practical usefulness of the proposed model. We fully agree with you that there is no WEF nexus model that can be directly applied across all spatial and temporal scales. Meaningful practical application of nexus models necessarily requires contextual development, often in close collaboration with local experts, stakeholders, and decision-makers.

This perspective aligns closely with the aim of our model. The generalisability of the proposed theoretical model is not intended to imply direct, context-free applicability or transferability. Rather, the generalisability of the model is contingent upon several key preconditions, which we clarify below.

- First, the model is designed as a general theoretical structure that comprehensively captures the core interactions within the WEF nexus, as well as the impacts of climate change, demographic dynamics, and socio-economic development on the nexus. This enables the model to investigate a wide range of WEF-related research questions.
- Second, the structure of the model—its variables, parameters, constraints, and interaction mechanisms—is not tied to any specific region. For example, a study area—whether a river basin, or an administrative region—can be divided into multiple spatial units ($s$), while the analytical focus remains on the interactions among these units. It is this structural level—rather than specific parameter values—that we consider transferable across contexts.
- Third, in practical applications, local experts, stakeholders, and researchers are expected to contextualize the theoretical model by selecting region-specific variables and parameters, as shown in Phase 2 of the roadmap in Fig.1. This includes identifying locally relevant consumer types (e.g., public vs. private consumption, rural vs. urban users), key firm activities (e.g., hydropower generation, rainwater harvesting, biofuel production), main resource commodities, and type of water sources (e.g., surface water, groundwater, aquifer water). Through this participatory and

context-specific process, the model will be tailored to the characteristics, constraints, and policy priorities of a given region.

To illustrate how the abstract theoretical model can be adapted and made transferable to specific contexts, and to demonstrate how it can address real-world challenges, we use the Beijing-Tianjin-Hebei (BTH) region as an example in the original manuscript. The BTH region is characterized by notable scarcity of natural resources, as well as unequal political and administrative rights affecting resource allocation among the three subregions. Hebei Province and Tianjin have often "sacrificed" to support Beijing, the capital of China. Since 2015, the central government has promoted coordinated development across the three regions. However, it remains unclear whether cross-regional cooperation has achieved a more balanced allocation of resources. Using this context as an example, our WEF nexus model naturally captures both cross-regional and cross-resource interactions, making it particularly suitable for practical applications. The model can provide policy-relevant insights to support coordinated resource management and inform decisions that promote more equitable WEF governance across regions.

Based on field research and publicly available statistics, we identify regionally important firm activities, consumer types, and resource commodities in BTH region, demonstrating how the theoretical model can be tailored to a concrete regional setting. This example is not intended as a full empirical application, but rather as a demonstration of how the model can be operationalized in a specific context, corresponding to Phases 1 and 2 of the roadmap shown in Fig. 1. We fully agree, however, that the discussion of the BTH region in the original manuscript remained at the level of theoretical structuring. Full practical implementation, including calibration, validation, and simulation, is beyond the scope of the present study and is being addressed in separate ongoing research.

To further enhance the applicability and usefulness of the model, in the revised manuscript, we plan to continue using the BTH case and add a dedicated section that explicitly advances to Phase 3 of the roadmap shown in Fig. 1, focusing on the selection of appropriate "normal years" and data requirements for the BTH region (the detailed dataset table will be supplemented; see response to comment 4 below for details). While this extension will not produce full empirical results, it will clarify the practical pathway from the theoretical framework to a context-specific application, without overstating the model's current empirical support.

3. **Comment:** How is the model proposed to be calibrated and validated? At the moment there are no results, yet no procedure for this aspect is considered.

   **Response:** We appreciate your comment on the calibration and validation of the model. We acknowledge that the manuscript did not clearly explain how the proposed model could be operationalized in practice, which weakened its perceived functionality. The primary intention of this paper is to develop and present the theoretical formulation of a WEF nexus optimization model. Yet, we are happy to provide more insights in the procedure that will be followed in an implementation step.

   First, we remark that the constraints of the model are material balances, which implies that the first step is to collect data on availability. For water, this includes rainfall, inflows from outside of the system and other "cells" within the system, retained water, and return flows of water from users. For energy, this includes data on energy generation from different sources and transport networks. For food, this includes data on agricultural production, imports from outside the system and trade

flows within the system. The same exercise needs to be done on the use side of the balances, considering consumer demand for the three commodities, outflows, and use by producers. In addition, technical parameters are collected from the three fields. For water, this includes runoff and percolation parameters; for energy, the maximum throughput in a system; and for food, parameters of agricultural and processing technologies. For each of the balances, an assessment will then be made of the gaps, and, depending on that assessment, either new data will be searched, or manual adjustments will be carried out based on qualitative information.

When all options to close the gaps are exhausted, step 2 of the calibration starts. This is exact calibration, using an optimization model where the objective is to minimize the gaps in the balances, and where constraints are given by bounds on parameter values that are assumed so far. The broadness of the bands around current values reflects the certainty on the current values: some parameters will be constrained to remain very close to initial values, while others will be allowed to adjust. After this step, there may still be small gaps; these will be accepted and kept fixed in simulations.

The final step is to bring the (now consistent) set of constraints under the optimization. For the utility functions used in the welfare optimum, standard techniques of estimating utility functions from demand data is used. The welfare weights are then adjusted to assure the baseline outcome is represented as the optimal outcome of the optimization given the balances.

**4. Comment:**

- Although data is mentioned in one line near the end of the paper, much more should be made of the actual feasibility of applying the theoretical model developed. It contains a vast amount of parameters, some being very specific. No note is made on where data could come from, data reliability, uncertainty, or the consequences to the model as a whole if large data are not available. Again, over 15 years experience tell me that often much compromise must be made in nexus modelling efforts as a result of data constraints. Reality is far from theoretical idealism.

- Line 290-295, but throughout the paper: where does one get all these coefficients and fractions? How are they verified and validated? Are they done on a per-case basis? In which case the model is not generally applicable…This again represents a major data challenge.

- Line 350: the technology-specific matrix à this again alludes to my comment on the data-intensiveness of the model and how feasible it would be to implement it in any meaningful way. More must be made of this issue.

**Response:** Thank you for this important comment regarding data challenges in the practical implementation of the theoretical model. We fully agree that meaningful application of the model requires careful consideration of data availability, reliability, and potential uncertainty, and often necessitates compromises that reflect real-world constraints.

We acknowledge that the model can be data-intensive, depending on the depth and scope of the planned study. For example, in our ongoing empirical study of the Beijing-Tianjin-Hebei region, we aim to capture all six mutual interactions among water, energy, and food resources, rather than focusing on isolated resource interactions. Besides, we aim to include detailed technical information of firm activities to reflect real-world operational processes. To this end, we have collected as much detailed

data and information as available for the BTH region. Our experience indicates that even at this high level of granularity, the model remains operationally feasible. This further suggests that simpler applications of the model, using a narrower set of parameters, would be even more manageable.

During the data collection process, we have indeed encountered some challenges, which directly correspond to your concerns. To explicitly address these issues, we plan the following revisions in the manuscript:

1) **Data table for illustration:** Following our response to comment 2 above, we will include a new Table 1 (shown below), using the BTH region—or more generally China—as an example. The table specifies the data inputs and parameters required to implement the proposed model. All variables and parameters listed in the table are based on official secondary data sources and published empirical studies. Where additional precision is desired, complementary information may be obtained through targeted field research and stakeholder engagement to further enhance the accuracy and regional specificity of the data. Note that this table is currently a draft; in the formal revision, it will be further refined.

2) **New discussion section on data challenges**: We will add a dedicated section before the conclusion to discuss practical data challenges in applying the model, mainly including:
   - **Data requirements and feasibility**: Implementation may require simplifications depending on data availability.
   - **Compromises in real-world applications:** Key parameters may need to be approximated or aggregated due to limited data.
   - **Data consistency and integration:** As the WEF nexus spans multiple sectors, datasets are collected from different resource sectors. Methods for integrating data across temporal and spatial resolutions will be discussed.
   - **Data resolution and temporal disaggregation:** While ideally production and consumption dynamics of the WEF nexus would be captured at a monthly or finer resolution, most available datasets are annual. Still, monthly dynamics can be approximated:
     - For variables insensitive to monthly variation, annual values can be evenly distributed across 12 months.
     - For monthly sensitive variables, monthly proportions can be derived from historical patterns in the literature and applied to a selected "normal year," with all assumptions clearly stated. These assumptions will be tested in subsequent sensitivity analyses.

We believe these revisions will substantially improve the clarity and credibility of the model's practical applicability while transparently acknowledging the limitations imposed by real-world data constraints.

Table 1. Overview of datasets

| Resource | | Variable | Temporal resolution | Spatial resolution | Corresponding variables/parameters/ indexes of the model | Data sources |
|---|---|---|---|---|---|---|
| Water | Demand | Total Water use
 • Agricultural
 • Industrial
 • Municipal
 • Household (rural and urban separately)
 • Environmental | Annual/Monthly | National/ Provincial (or municipality)/ District | $d^i, e^j$ | Local Statistical Yearbooks; Local Water Resources Bulletin; Local Bureau of Statistics; Literature (e.g., Long, D., Yang, W., Scanlon, B.R. et al. South-to-North Water Diversion stabilizing Beijing's groundwater levels. Nat Commun 11, 3665 (2020). https://doi.org/10.1038/s41467-020-17428-6) |
| | | Water use by sector
 • Ming and washing of coal
 • Production and distribution of electricity power and heat power
 • Food processing
 • Production and distribution of gas
 • …… | Annual | National/Provincial (or municipality)/ | $e^j$ | Local Statistical Yearbooks; Local Water Resources Bulletin; Local Bureau of Statistics |
| | | Groundwater use
 • Agricultural
 • Industrial
 • Domestic
 • Environmental | Monthly | Provincial (or municipality | $d^i, e^j$ | Literature (e.g., Long, D., Yang, W., Scanlon, B.R. et al. South-to-North Water Diversion stabilizing Beijing's groundwater levels. Nat Commun 11, 3665 (2020). https://doi.org/10.1038/s41467-020-17428-6) |
| | | …. | | | | |
| | Supply | Total Water supply
 • Surface water | Annual/Monthly | National/Provincial (or | $e^j, l$ | Local Statistical Yearbooks; Local Water Resources Bulletin; Local |

| | | | | | |
|---|---|---|---|---|---|
| | | • Groundwater
• Transfer water
• Reclaimed water | | municipality)/
District | | Bureau of Statistics; Literature (e.g., Long, D., Yang, W., Scanlon, B.R. et al. South-to-North Water Diversion stabilizing Beijing's groundwater levels. Nat Commun 11, 3665 (2020). https://doi.org/10.1038/s41467-020-17428-6) |
| | | Precipitation | Annual/Monthly/Daily/Hourly | National/Provincial (or municipality)/District/County/Grid | $b$ | Local Statistical Yearbooks; Local Water Resources Bulletin; Local Bureau of Statistics; China Meteorological Data Center, Resource and Environmental Science Data Center, Chinese Academy of Sciences |
| | | … | | | | |
| | Quality | Volume of Wastewater Discharged
• Industrial Source
• Residential Source
• Centralized Treatment Facilities | Annual | National/Provincial (or municipality) | $e^j$ | China Environmental Statistic Yearbook; Local Statistical Yearbooks; Local Water Resources Bulletin; China Urban Construction Statistical Yearbook |
| | | COD of Wastewater Discharged
• Industrial Source
• Residential Source
• Agricultural Source
• Centralized Treatment Facilities | Annual | National/Provincial (or municipality) | $e^j$ | China Environmental Statistic Yearbook; Local Statistical Yearbooks; Local Water Resources Bulletin; China Urban Construction Statistical Yearbook |
| | | Ammonia and Nitrogen of Wastewater Discharged
• Industrial Source
• Residential Source
• Agricultural Source
• Centralized Treatment Facilities | Annual | National/Provincial (or municipality) | $e^j$ | China Environmental Statistic Yearbook; Local Statistical Yearbooks; Local Water Resources Bulletin; China Urban Construction Statistical Yearbook |

| | | Total Wastewater Discharged
• Total Nitrogen
• Total Phosphorus
• Petroleum
• Mercury
• Cadmium
• Arsenic
• … | Annual | National/Provincial (or municipality) | $h$ | China Environmental Statistic Yearbook; Local Statistical Yearbooks; Local Water Resources Bulletin; China Urban Construction Statistical Yearbook |
|---|---|---|---|---|---|---|
| | | Water temperature | Real-time, monitoring | Specific monitoring sites | $c$ | Local Water Authority |
| | | Wastewater Treatment capacity | Annual | National/Provincial (or municipality) | $e^j$ | China Urban Construction Statistical Yearbooks; Local Statistical Yearbooks |
| | | … | | | | |
| | Stock | Total Water Resources
• Surface Water Resources
• Groundwater Resources
• Non-overlapping amount of groundwater and surface water resources | Annual/Monthly | National/Provincial (or municipality) | $\tilde{x}$ | China Environmental Statistic Yearbook; Local Statistical Yearbooks; Local Water Resources Bulletin |
| | | Transfer water between regions
• Inflow volume
• Outflow volume | Annual | National/Provincial (or municipality) | $e^j$ , $b$ | Local Statistical Yearbooks; Local Resources Bulletin; Haihe River Basin Water Resources Bulletin |
| | | … | | | | |
| | Others | Runoff coefficient | Annual/Real-time monitoring | National/Provincial (or municipality) | $A$ | Local Water Resources Bulletin; Local Water Authority |
| | | Percolation parameter | Annual | National/Provincial (or municipality) | $A$ | Literature |

| | | | | | | |
|---|---|---|---|---|---|---|
| | | Effective Utilization Coefficient of Farmland Irrigation Water | Annual | National/Provincial (or municipality) | $B^j$ | China Water Resources Bulletin |
| | | … | | | | |
| Energy | Supply | Energy production
• Thermal power
• Hydropower
• Nuclear power
• Wind power
• Solar power
• … | Annual/Monthly | National/Provincial (or municipality) | $e^j, b$ | China Energy Statistical Yearbook; State Power |
| | | Fuel for heating and cooking for urban residents and rural residents separately
• Firewood
• Coal
• Liquefied Petroleum Gas
• Coal Gas
• Natural Gas
• Electricity
• Fuel Oil
• Methane
• … | Annual | National/Provincial (or municipality) | $d^i$ | Local Survey Yearbook |
| | | Energy balance tables for
• Raw Coal
• Coke
• Crude Oil
• Gasoline
• Kerosene
• Diesel Oil
• Liquefied Petroleum Gas
• Natural Gas | Annual | National/Provincial (or municipality) | $B^j, e^j$ | China Energy Statistical Yearbook; Local Statistical Yearbook |

| | | | | | | |
|---|---|---|---|---|---|---|
| | | • Liquefied Natural Gas
• Heat
• Electricity | | | | |
| | | Inter-regional power exchange | Annual | Provincial (or municipality) | $e^j$, $b$ | Literature |
| | | Industrial production
• Chemical fertilizer | Annual | National/Provincial (or municipality) | $e^j$ | China Statistical Yearbook; Local Statistical Yearbook |
| | | … | | | | |
| | Demand | Total Electricity Consumption
• First Industry
• Second Industry
• Third Industry
• Households | Annual | National/Provincial (or municipality) | $e^j$ | China Statistical Yearbook; Local Statistical Yearbook |
| | | Energy consumption by sector
• Agriculture, forestry, animal husbandry, and fishery
• Mining
• Mining and washing of coal
• Extraction of petroleum and natural gas
• Processing of food from agricultural products
• Manufacture of foods | Annual | National/Provincial (or municipality) | $e^j$ | China Statistical Yearbook; Local Statistical Yearbook |
| | | Power of agricultural machinery by energy type
• Electricity
• Diesel
• … | Annual | National/Provincial (or municipality) | $e^j$ | National Statistical Yearbooks; Local Statistical Yearbooks; |
| | | Electricity consumption per capital
• Rural households
• Rural households | Monthly | National/Provincial (or municipality)/ City/1km×1km Grid | $d^i$ | China Energy Statistical Yearbook; Literature (e.g., Yan, X., Huang, Z., Ren, S., Yin, G., & Qi, J. (2024). Monthly electricity consumption data at 1 km × 1 km |

| | | | | | | grid for 280 cities in China from 2012 to 2019. Scientific Data, 11(1), 877. https://doi.org/10.1038/s41597-024-03684-4; Du, M., Ruan, J., Zhang, L., Niu, M., Zhang, Z., Xia, L., Qian, S., & Chen, C. (2024). China's local-level monthly residential electricity power consumption monitoring. Applied Energy, 359, 122658. https://doi.org/10.1016/j.apenergy.2024.122658) |
|---|---|---|---|---|---|---|
| | | … | | | | |
| | Others | Conversion factors from physical units to coal equivalent | Annual | National | | China Energy Statistical Yearbook |
| | | Efficiency of Energy Transformation | Annual | National | $B^j$ | China Energy Statistical Yearbook |
| | | Loss of energy in transport | Annual | National | $B^j, e^j$ | China Energy Statistical Yearbook; Literature |
| | | …. | | | | |
| Food | Supply | Sown Area and output for
• Rice
• Wheat
• Corn
• Soybean
• Tubers
• Oil-bearing Crops
• Cotton
• Vegetables
• Fruits
…. | Annual/Quarter | National/Provincial (or municipality)/District/County | $e^j$ | National Statistical Yearbooks; Local Statistical Yearbooks; Local Bureau of Statistics |

| | | Stock number of
• Hogs
• Cattle and Buffaloes
• Sheep and Goats
• Poultry | Annual/Quarter | National/Provincial (or municipality)/District/County | $e^j$ | National Statistical Yearbooks; Local Statistical Yearbooks; Local Bureau of Statistics |
|---|---|---|---|---|---|---|
| | | Output of
• Pork
• Beef
• Mutton
• Poultry
• Poultry Eggs
• Milk | Annual | National/Provincial (or municipality)/District/County | $e^j$ | National Statistical Yearbooks; Local Statistical Yearbooks; Local Bureau of Statistics |
| | | International import
• Maize
• Corn
• Pork
• … | Annual/Monthly | | $b$ | Ministry of Agriculture and Rural Affairs of the People's Republic of China |
| | | Intermediate consumption of main crops and livestock products, including the cost of fuel, forage, seed, water, electricity | Annual | National/Provincial (or municipality) | $B^j$ | China Rural Statistical Yearbook; Local Rural Statistical Yearbook |
| | | Consumption of chemical fertilizer and pesticide | Annual | National/Provincial (or municipality) | $e^j$ | China Rural Statistical Yearbook; Local Rural Statistical Yearbook |
| | | Comprehensive Utilization Rate of Livestock and Poultry Manure | Annual | National/Provincial (or municipality) | $B^j$ | China Rural Statistical Yearbook; Local Rural Statistical Yearbook |
| | | Comprehensive Utilization Rate of Crop Straw | Annual | National/Provincial (or municipality) | $B^j$ | China Rural Statistical Yearbook; Local Rural Statistical Yearbook |
| | | … | | | | |

| | | | | | | |
|---|---|---|---|---|---|---|
| | Demand | Per Capita Consumption of Rural households, and urban households separately:
• Grain
• Cereals
• Tuber
• Beans and Products
• Edible Oil and Fats
• Vegetables and Edible Fungi
• Pork
• Beef and Mutton
• Poultry and Poultry Products
• Eggs and Related Products
• Milk and Dairy Products | Annual | National/Provincial (or municipality) | $d^i, m$ | China Statistical Yearbook; Local Statistical Yearbook |
| | | International export
• Maize
• Corn
• Pork
• … | Annual/Monthly | | $b$ | Ministry of Agriculture and Rural Affairs of the People's Republic of China |
| | | … | | | | |
| | Risk | Total area affected of farm crops, by region, by disaster types:
• Drought
• Flood and geological disaster
• Wing
• Hyphen disaster
• Low temperature freezing and snow disaster
• … | Annual | National/Provincial (or municipality) | $e^j$ | China Rural Statistical Yearbook; Local Rural Statistical Yearbook |
| | | … | | | | |
| | Price | Price for
• Wheat
• Corn | Daily, Weekly, | National/Provincial (or municipality) | | Agricultural Product Price Survey Yearbook; Ministry of Agriculture |

| | | | | | |
|---|---|---|---|---|---|
| | | • Soybean
• Cotton
• … | Monthly,
Annual | | | and Rural Affairs of the People's Republic of China |
| | | … | | | | |
| | Others | Parameters of agricultural and food processing technologies | Annual | National/Provincial (or municipality) | $B^j$ | China Rural Statistical Yearbook; Local Rural Statistical Yearbook; Literature |
| | | … | | | | |
| Meteorological | | • Average Temperature
• Highest Temperature
• Lowest Temperature
• Hours of Sunshine
• Average Wind Speed
• Average Relative Humidity
• … | Annual/Monthly/Daily/Hourly | National/Provincial (or municipality)/District/County/Grid | | Local Statistical Yearbooks; China Meteorological Data Center; Resource and Environmental Science Data Center; Chinese Academy of Sciences |
| | | … | | | | |
| Socio-economic | | Permanent population
• Male population
• Female population
• Urban population
• Rural population | Annual | National/Provincial (or municipality)/District/County | $i$ | Local Statistical Yearbooks |
| | | … | | | | |

**Response to Specific Comments**

1.  **Comment:** The introduction lacks any real mention of the criticisms levelled at "the nexus". There is a wide body of literature on this topic, which should be included to give a more balanced perspective.

    **Response:** Thank you for this insightful comment. We fully agree that there is a substantial body of literature discussing criticisms of the WEF nexus. To provide a more balanced perspective, we will ensure to include a more balanced review of key criticisms and debates surrounding the nexus concept in the Introduction. This will help clarify the key criticisms in the literature and situate our theoretical model within this broader discussion.

2.  **Comment:** The term 'ideal model' is used throughout the paper yet never properly defined. This should be addressed.

    **Response:** Thank you for this comment. We agree with you that the term "ideal model" may be misleading, as there is no truly perfect model. We will revise the manuscript to avoid using this term and clarify our intended meaning wherever it appears.

3.  **Comment:** Line 34: "a model should achieve resource security…". No model can achieve this. It can merely suggest possible pathways towards it. Please rephrase.

    **Response:** Thank you for this comment. We agree with the you that a model itself cannot achieve resource security. We will rephrase the statement to clarify that the model is intended to provide insights that could help guide decisions towards improving resource security. We will change the "achieve resource security to "enhance resource security" throughout the manuscript.

4.  **Comment:** Line 40: water quality in addition to temperature and heavy metals has been studied in a WEF nexus context. See Amorocho-Daza H., Sušnik J., Slinger J.H., van der Zaag P. 2026. A participatory system dynamics approach to assess transboundary nutrient pollution: modelling the water-energy-food-ecosystems nexus in the Lielupe River Basin, Lithuania and Latvia. Ecological Modelling. 513: 111417.

    **Response:** Thank you for bringing this paper to our attention. After reviewing it in detail, we note that the study uses nitrogen concentration as the sole proxy for water quality. Regarding temperature, it only appears in Fig. 3D (a) as part of the climate impact on water; in this context, the temperature likely refers to air temperature rather than water temperature. This paper actually supports our original claim on line 39: "Specialized water models have integrated water temperature and heavy metals, especially in rivers, but these are absent in WEF nexus studies."

    Looking beyond the WEF nexus field, in hydrology, the importance of water temperature and heavy metals as water quality indicators—and their incorporation into models—has been widely recognized (Ficklin et al., 2023; Ouellet et al., 2020; Van Vliet et al., 2023). However, their integration into WEF nexus studies remains limited, even in the study published in 2026 you referenced. These water quality indicators play a critical role in WEF nexus interactions. For instance, thermoelectric power plants depend on adequate cooling water, while agricultural productivity is sensitive to water temperature and contamination. This highlights the importance of explicitly incorporating water temperature and heavy metals into WEF nexus modelling, particularly in the context of increasingly frequent extreme heat events.

**Reference:**

- Ficklin, D. L., Hannah, D. M., Wanders, N., Dugdale, S. J., England, J., Klaus, J., Kelleher, C., Khamis, K., & Charlton, M. B. (2023). Rethinking river water temperature in a changing, human-dominated world. Nature Water, 1(2), 125–128. https://doi.org/10.1038/s44221-023-00027-2

- Ouellet, V., St-Hilaire, A., Dugdale, S. J., Hannah, D. M., Krause, S., & Proulx-Ouellet, S. (2020). River temperature research and practice: Recent challenges and emerging opportunities for managing thermal habitat conditions in stream ecosystems. Science of The Total Environment, 736, 139679. https://doi.org/10.1016/j.scitotenv.2020.139679

- Van Vliet, M. T. H., Thorslund, J., Strokal, M., Hofstra, N., Flörke, M., Ehalt Macedo, H., Nkwasa, A., Tang, T., Kaushal, S. S., Kumar, R., Van Griensven, A., Bouwman, L., & Mosley, L. M. (2023). Global river water quality under climate change and hydroclimatic extremes. Nature Reviews Earth & Environment, 4(10), 687–702. https://doi.org/10.1038/s43017-023-00472-3

5. **Comment:** Line 65: the fact that the model developed here builds on a model developed for the Jordan River seems to be in contradiction of it being generally applicable. This discrepancy needs to be addressed. For example, how could such a specific model be extended to be widely applicable across time, space, and contexts? Also, the JRB model is said to exclusively represent water, thus contradicting its ability to model nexus interactions. Does taking a water-centric model represent a good starting place for an alleged 'ideal' nexus model? If so why? I feel this argument is not yet made to an adequate degree.

   **Response:** Thank you for raising this thoughtful concern. We agree that, without clarification, building on a model originally developed for the Jordan River Basin may appear to be in tension with the claim of general applicability. We therefore clarify below why the JRB model serves as a suitable starting point, rather than a constraint, for the proposed WEF nexus model.

   First, following our adjustments and reinterpretation of the JRB model, water is now treated as one resource commodity among others, rather than the organizing centre of the system. By explicitly extending the model to include food and energy as parallel resource commodities, the model now captures cross-resource interactions in a balanced way, rather than being water-centric. This ensures that the JRB model's original strengths—its ability to capture interactions among resources—are preserved and generalized.

   Second, the underlying structure of the JRB model is not specific to the Jordan River Basin. Its mathematical formulation and mechanisms are sufficiently general to support the six objectives proposed in our manuscript. This general structure allows the model to be adapted across different regions, time periods, and socio-economic contexts, rather than being tied to the JRB only.

   Taken together with the two points above and our previous response to your general comment 2 regarding the generalisability of the model, the presented model can leverage the strengths of a well-established water-centric model while providing a balanced representation of resources and their interactions within the WEF nexus. We will revise the manuscript to make these points more explicit.

6. **Comment:** There is multiple mention on 'pure water', but this is not defined. Even 'pure water' contains nutrients, minerals, suspended material.

**Response:** Thank you for this comment. We agree that, in the physical world, even so-called "pure water" contains minerals, nutrients, and suspended materials. In our model, the water that is present at any location is a mixture of analytically separated flows. One of the flows is pure $H_2O$. The reason for modelling water in this way is that we can maintain mass balances for the separate flows, allowing also for potentially different speeds if items are not fully soluble.

Specifically, in the model, "pure water" refers to water without the presence of explicitly modelled water pollutants. This formulation allows us to separately represent and track different pollutants of interest (e.g., nitrogen, lead) by defining individual pollutant-specific flows with corresponding maximum concentrations, shown in (a) of Fig. 2. These flows are then combined to represent the mixed water quality actually received by users, as illustrated in (b) of Fig. 2. This design reflects real-world water use more realistically while maintaining analytical clarity.

To avoid ambiguity, we will revise the manuscript to explicitly define "pure water" upon its first occurrence as a theoretical reference state representing water without pollutants relevant to the specific processes modelled.

[Figure]

Figure 2. (a) Representing pure water ($H_2O$) and pollutants as separate flows, (b) Representing the blending of pure water ($H_2O$) and pollutants.

7. **Comment:** Food quality à what does this actually measure/mean in practice? What is/are the variables being tracked here? Food access? Cost? Nutritional value? Calories? Against what benchmark(s)? As with other variables, this need expanding to have any practical meaning. How is food quality (however measured) affected in the model?

**Response:** Thank you for raising this question and suggesting various measurements of food quality. In this study, food quality is defined from the resource-interaction perspective, which is central to the WEF nexus. We distinguish three types of food quality, corresponding to different resource-use pathways in the nexus:

1) Food for human consumption ($h_1$) – representing direct human food use; consumption ultimately generates return flows such as manure, capturing *food for food* interaction.
2) Food for animal feed ($h_2$) – representing food converted into animal products for human consumption, capturing *food for food* interaction.
3) Food for energy conversion ($h_3$) – representing crops and crop residues used for energy generation (e.g., biofuel), capturing *food for energy* interactions.

This distinction serves several practical purposes:

- It reflects real-world differences in food use, where higher-quality food for human consumption cannot be fully substituted by feed or energy crops ($h_1 > h_2 > h_3$).
- It captures trade-offs and competition across food uses based on quality differences.
- Given the temporal dimension of the model, it can account for food storage and transportation, and the related food spoilage of these activities.

We will revise the manuscript to more clearly explain how food quality is affected by and feeds back into the nexus.

8. **Comment:** Line 265: what is 'homogeneous energy'? And as with the food comment, what is meant by energy quality? Is it access? Hours of access? Reliability? Fuel source? Cooking fuel? Source of power? How is it affected in the model?

**Response:** Thank you for this comment. In practice, energy quality, especially for electricity, is often measured by factors such as frequency of outages, voltage fluctuations, or hours of full-service provision per day (Meeks et al., 2023). While these factors are important in engineering contexts, they do not directly interact with other resources in the WEF nexus.

To maintain analytical clarity and focus on resource interactions, we therefore assume that energy is homogeneous in quality—i.e., there is no variation in reliability, access, or performance within a given energy flow. Different energy types (e.g., coal, natural gas, electricity) are distinguished in the model via the resource commodity variable $m$, but within each type, the model treats energy flows as uniform in quality. This assumption allows us to capture the interactions among resources without conflating them with service-quality metrics that are external to the nexus.

We will revise the manuscript to explicitly define "homogeneous energy" and clarify our assumptions regarding energy quality upon first introduction, to avoid ambiguity for the reader.

**Reference:**

Meeks, R. C., Omuraliev, A., Isaev, R., & Wang, Z. (2023). Impacts of electricity quality improvements: Experimental evidence on infrastructure investments. *Journal of Environmental Economics and Management*, *120*, 102838. https://doi.org/10.1016/j.jeem.2023.102838

9. **Comment:** Line 301: mention is made here of 'cells'. So is the model an agent-based model/cellular automata? If so, this needs to be explicitly mentioned.

**Response:** Thank you for this thoughtful comment. In our model, a cell represents the minimum analytical unit, defined by the combination of six dimensions: $h$, $s$, $t$, $l$, $m$, and $c$. Within each cell, a resource balance model is solved:

$$\sum_i d^i + \sum_j e^j + \tilde{x} = A\tilde{x} + \sum_i B^{0i} d^i + \sum_j B^j e^j + b$$

This formulation allows the model to capture two types of resource interactions: (1) within-cell interactions, reflecting resource uses in the same spatial-temporal unit, and (2) cross-cell interactions, representing flows between different cells.

Importantly, the flexible nature of cells—where the sets of $h$, $s$, $t$, $l$, $m$, and $c$ can be tailored to local conditions—contributes directly to the generalisability and transferability of the model. By adjusting cell-specific inputs based on local data, the model can be applied across different spatial and temporal scales, as well as across diverse geographic and socio-economic contexts.

We will clarify in the revised manuscript that, while the model uses a "cellular" structure, it is not an agent-based or cellular automata model; the term "cell" is used purely as a modelling unit to structure resources and interactions systematically.

10. **Comment:** Line 325: What about food waste, energy waste? Are these considered? Food and energy can be lost from a system (degraded food, waste heat, transmission losses).

    **Response:** Thank you for raising this important point. Food and energy waste are indeed crucial considerations in real-world WEF nexus systems, and both are considered in our model. In our model, resource waste arises from human consumption, while resource loss arises from production, processing, and transportation activities.

    For food, the supply for human consumption is divided into two components: (i) the portion that is effectively consumed by humans is absorbed and converted into residuals such as manure, which are modelled as a return flow from human consumption, and (ii) the portion of food that is supplied but not consumed is treated as food waste. A share of this food waste can be recovered—such as through conversion to energy or reuse as animal feed—and re-enters the nexus system as a new resource, represented by $B^{0i}d^i$. Here, $B^{0i}$ is a transformation matrix capturing the degree of reuse of food waste for a given region and time period. This design allows us to account for context-specific and temporal variations in food waste, such as increases during the Spring Festival. In addition, food loss and degradation occurring during production, processing, and transportation activities are captured by the corresponding transformation matrix $B^j$, in which outputs are lower than inputs.

    Energy waste is treated differently due to its distinct physical nature. Once energy is consumed, it generally does not generate a tangible return flow that can be reused within the WEF nexus. For example, fossil fuel consumption results in waste heat and greenhouse gas emissions that exit the system. Accordingly, no recovery term $B^{0i}d^i$ is defined for energy commodities, consistent with the discussion in line 296 of the manuscript. Losses associated with energy production, conversion, and transmission, arising from firm activities such as power generation and energy transport, are modelled through transformation matrices $B^j$ with outputs lower than inputs.

    By distinguishing between recoverable and non-recoverable waste and losses, the model provides a coherent representation of waste generation, loss, and potential reuse within the WEF nexus. We will further clarify this mechanism and expand the discussion of food and energy waste in the revised manuscript to improve transparency and interpretation.

11. **Comment:** Line 332: 'no resources will disappear'. This is unrealistic and should be addressed.

    **Response:** Thank you for pointing this out. We agree that the original wording, "no resources will disappear", is potentially misleading when interpreted in a strict physical sense. Our intention was not to suggest that resources are conserved without loss, but rather to emphasize that, consistent with the principle of resource circularity, resources are explicitly tracked within the modelling

framework as they are transferred across locations or transformed into other forms or qualities (e.g. loss, evaporation or degradation), rather than being implicitly ignored.

To avoid any potential confusion, we will revise the wording throughout the manuscript to reflect this distinction more clearly and to ensure consistency with the physical interpretation of resource losses.

12. **Comment:**

- Line 345: e.g. reduced water quality and elevated temperature. How are these changes estimated in the model? There is a lot of ambiguity here. As mentioned above, a conceptual figure showing model interactions would significantly help. I know there are some figures like this in the appendix, but I feel some more specific examples to illustrate key processes should be included.

- Following from the above comment – a diagram showing causal impacts in the model would be of great value to the reader to understand nexus interactions.

**Response:** Thank you for this helpful comment and suggestion. At present, how the changes in water quality and temperature are examined by the model are described conceptually in the manuscript text. We agree that this textual explanation alone may lead to ambiguity and make it difficult for readers to clearly understand the underlying mechanisms. Following your suggestion, in the revision stage we will improve the clarity of this part by adding more explicit figures to illustrate the key interactions and processes of key firm activities.

13. **Comment:** The social welfare metric is relatively poor. There are far better indicators of social progress and benefit. Why was this specific metric chosen over others? For example nothing is mentioned of equity, access, distribution, or the sustainability of consumption of finite resources.

**Response:** Thank you for raising this substantive concern. We fully agree that social progress and benefit can be measured using a wide range of indicators. Many of these indicators have indeed been widely discussed and applied in nexus studies (Stone et al., 2023).

In this study, however, our objective is not to provide an exhaustive representation of societal dimensions, but rather to introduce a parsimonious yet operational social welfare metric that can be endogenously integrated into an optimization model. The choice of the metric is therefore guided by feasibility, internal consistency, and its ability to explicitly represent equity. Specifically, equity considerations are incorporated by:

- **Intergenerational equity**, captured through the time-discount factor $\beta$, which assigns value to utility derived from remaining resources available to future generations and thus reflects the sustainability of consumption over time.

- **Intragenerational equity**, represented by heterogeneous welfare weights $\alpha_i$, assigned to different consumer groups (e.g., urban vs. rural households). These weights allow the model to explicitly account for distributional preferences and unequal welfare impacts across population groups.

The decision to focus on these two dimensions is motivated by identified research gaps:

- Much of the quantitative WEF nexus studies prioritises efficiency, emissions reduction, or system resilience, while the explicit quantification of equity remains relatively limited. Our inclusion of these two social equity dimensions therefore intend to contribute to a more diverse measurement of social considerations, rather than to duplicate social indicators that have already been extensively used in the literature.

- Most existing nexus models treat the population as a homogeneous aggregate or focus primarily on urban users (Zhang et al., 2024), which restricts their ability to analyse distributional outcomes, particularly in low-income country contexts where rural-urban disparities are substantial.

By embedding intergenerational and intragenerational equity directly into the social welfare function, our model integrates societal considerations with economic and environmental considerations, rather than as parallel or ex-post indicators. This design enhances analytical coherence and enables systematic trade-off analysis within a unified modelling framework.

To address your concern more clearly, we will revise the text to (i) more explicitly justify the choice of the social welfare metric, and (ii) openly discuss the potential for incorporating additional social indicators.

**Reference:**

- Stone, T. F., Dickey, L. C., Summers, H., Thompson, J. R., Rehmann, C. R., Zimmerman, E., & Tyndall, J. (2023). A systematic review of social equity in FEWS analyses. Frontiers in Environmental Science, 11, 1028306. https://doi.org/10.3389/fenvs.2023.1028306

- Pengpeng Zhang, Lixiao Zhang, Yan Hao, Ming Xu, Mingyue Pang, Changbo Wang, Aidong Yang, Alexey Voinov, Food–energy–water nexus optimization brings substantial reduction of urban resource consumption and greenhouse gas emissions, PNAS Nexus, Volume 3, Issue 2, February 2024, pgae028, https://doi.org/10.1093/pnasnexus/pgae028

14. **Comment:** Appendix C: how are changes in crop yield estimated?

**Response:** Thank you for the comment. We assume that you refer to changes in crop yield resulting from changes in water availability and quality. For this, we rely on earlier research done by one of the authors on Chinese agriculture, where these impacts have been determined based on existing empirical literature and agronomic parameters (Van Wesenbeeck et al., 2021, Yu et al., 2026).

**References:**
- Van Wesenbeeck, C. F. A., Keyzer, M. A., Van Veen, W. C. M., & Qiu, H. (2021). Can China's overuse of fertilizer be reduced without threatening food security and farm incomes? *Agricultural Systems*, *190*, 103093. https://doi.org/10.1016/j.agsy.2021.103093
- Le Yu, Van Wesenbeeck, Van Veen, W. C. M. (2026). Farmer choices under climate change in China: a regional analysis of food versus feed production in 2050, *working paper*.

15. **Comment:** Water-for-energy, line 598: hydropower does consume water – rather a lot. See e.g. https://www.sciencedirect.com/science/article/pii/S0960148116306176,

https://onlinelibrary.wiley.com/doi/full/10.1002/gch2.201600018, and
https://hess.copernicus.org/articles/17/3983/2013/ as just a few examples

**Response:** Thank you for the correction. What we originally intended to convey is that the generation of hydroelectricity itself relies on the gravitational potential of water and does not directly consume water in the way that, for example, irrigation withdraws water for crop growth. Instead, the main water losses associated with hydropower arise indirectly from increased surface evaporation due to the presence of reservoirs, a phenomenon that would similarly occur in other large open water bodies such as lakes, consistent with the studies you suggested (Bakken et al., 2017).

In our model, evaporation from open water surfaces, including reservoirs, is explicitly accounted for in the river layer, separate from the firm activity layer (which the hydropower generation belongs to). Specifically, in the resource balance model, the net exogenous availability $b$ represents precipitation minus immediate evaporation plus inflow from outside the system, thereby capturing the water losses due to surface evaporation associated with reservoirs. Hydropower generation itself is represented as one firm activity $j$ within the firm activity layer $l$, where electricity generation does not directly consume water. In this way, water consumption due to hydropower is separated into two components in our model: (i) direct consumption by the activity itself, which is assumed as zero, and (ii) evaporation from reservoirs.

We will revise the manuscript to clarify this distinction and to ensure that readers fully understand how hydropower-related water consumption is represented in the model.

**References:**

Bakken, T. H., Killingtveit, Å., & Alfredsen, K. (2017). The Water Footprint of Hydropower Production—State of the Art and Methodological Challenges. Global Challenges, 1(5), 1600018. https://doi.org/10.1002/gch2.201600018

We hope that these planned revisions adequately address your concerns. The revised manuscript will more clearly position the proposed model as a theoretical foundation for future context-specific WEF nexus applications. We are grateful for your time and expert guidance, which have substantially helped us strengthen the manuscript.